# Categorical Flow Matching on Statistical Manifolds

**Chaoran Cheng**[*]
University of Illinois Urbana-Champaign
chaoran7@illinois.edu

**Jiahan Li**[*]
Peking University
lijiahanypc@pku.edu.cn

**Jian Peng**
University of Illinois Urbana-Champaign
jianpeng@illinois.edu

**Ge Liu**
University of Illinois Urbana-Champaign
geliu@illinois.edu

## Abstract

We introduce Statistical Flow Matching (SFM), a novel and mathematically rigorous flow-matching framework on the manifold of parameterized probability measures inspired by the results from information geometry. We demonstrate the effectiveness of our method on the discrete generation problem by instantiating SFM on the manifold of categorical distributions whose geometric properties remain unexplored in previous discrete generative models. Utilizing the Fisher information metric, we equip the manifold with a Riemannian structure whose intrinsic geometries are effectively leveraged by following the shortest paths of geodesics. We develop an efficient training and sampling algorithm that overcomes numerical stability issues with a diffeomorphism between manifolds. Our distinctive geometric perspective of statistical manifolds allows us to apply optimal transport during training and interpret SFM as following the steepest direction of the natural gradient. Unlike previous models that rely on variational bounds for likelihood estimation, SFM enjoys the exact likelihood calculation for arbitrary probability measures. We manifest that SFM can learn more complex patterns on the statistical manifold where existing models often fail due to strong prior assumptions. Comprehensive experiments on real-world generative tasks ranging from image, text to biological domains further demonstrate that SFM achieves higher sampling quality and likelihood than other discrete diffusion or flow-based models. Our code is available at https://github.com/ccr-cheng/statistical-flow-matching.

## 1 Introduction

Recently, conditional flow matching (CFM) models [38] have achieved remarkable success in various generative domains including image generation [38, 17, 29], molecule [59, 58, 31] and protein design [68, 10, 36], and sequence generation [60, 39, 12]. While attempts to generalize CFM and diffusion models to discrete categorical data have been made, they typically exert ad hoc assumptions on the structure of the discrete distribution. One group of work relies on stochastic jumps of Markov chains in either the discrete-time [6, 39, 1] or continuous-time setting [11, 55, 12] that discards the continuous nature of the underlying categorical distributions. Other work directly works with the probability simplex [7, 60] or the corresponding logit space [28, 40, 24] with potentially imperfect assumptions that fail to capture the underlying true geometry of the statistical manifold. Furthermore, likelihood is often approximated by variational bounds in previous discrete generative models due to the lack of tractable exact likelihood.

---

[*]Equal contribution.

38th Conference on Neural Information Processing Systems (NeurIPS 2024).

We propose to incorporate the intrinsic geometry of the *statistical manifold* by viewing categorical data as points on the statistical manifold of categorical distributions. Inspired by the mathematical results from information theory, we utilize the Fisher information metric [51] to naturally equip such a manifold with a Riemannian structure and develop an efficient generative training scheme for learning the vector fields without stability issues. We summarize our contributions as the following:

(1) We propose *Statistical Flow Matching* (SFM), a novel and mathematically rigorous generative framework on the manifold of parameterized probability measures. SFM does not pose any prior assumptions on the statistical manifold but instead deduces its intrinsic geometry via mathematical tools. To tackle the discrete generation problem, we instantiate SFM on the manifold of categorical distributions. We deduce closed-form exponential and logarithm maps and develop an efficient flow-matching training algorithm that avoids numerical issues by leveraging a diffeomorphism between manifolds. SFM effectively leverages the intrinsic geometric properties by following the shortest paths of geodesics between the noise and target distributions on the statistical manifold.

(2) Our distinctive geometric perspective of the statistical manifold allows us to further apply optimal transport during training and derive tractable exact likelihood for any given sample of probability measure, both of which are unachievable for most existing methods. We also introduce new theoretical insights by establishing connections among Riemannian flow matching, information geometry, and natural gradient descent, which allows us to interpret SFM as following the steepest descent of the *natural gradient* from the optimization angle.

(3) We demonstrated with a toy example on simplex that SFM can learn more complex patterns on the statistical manifold where existing models often fail due to impromptu prior assumptions. We further conducted extensive experiments on diverse real-world discrete generation tasks involving computer vision, natural language processing, and bioinformatics. SFM consistently outperformed existing diffusion or flow-based models and also achieved comparable results with autoregressive models on character-level generation.

## 2 Preliminary

### 2.1 Information Geometry

In this work, we are interested in learning a parameterized family of probability measures. It is known from information theory that all probability measures over the sample space form the structure known as *statistical manifold*. Mathematically, consider probability densities $p = \frac{d\mu}{d\nu} : \mathcal{X} \to \mathbb{R}$ defined by the Radon-Nikodym derivative where $\mu$ is a probability measure on the sample space $\mathcal{X}$ and $\nu$ is the reference measure on $\mathcal{X}$. Suppose the statistical manifold $\mathcal{P} = \mathcal{P}(\mathcal{X}) = \{p : \int_{\mathcal{X}} d\mu = \int_{\mathcal{X}} p(x; \theta) \, d\nu = 1\}$ is parameterized by $\theta = (\theta_1, \theta_2, \ldots, \theta_n) \in \Theta$, this parameterization naturally provides a coordinate system for $\mathcal{P}$ on which each point is a probability measure $\mu$ with the corresponding probability density function $p(x; \theta)$. The *Fisher information metric* is defined as

$$g_{jk}(\theta) = \mathbb{E}_X \left[ \frac{\partial \log p(X; \theta)}{\partial \theta_j} \frac{\partial \log p(X; \theta)}{\partial \theta_k} \right] = \int_{\mathcal{X}} \frac{\partial \log p(x; \theta)}{\partial \theta_j} \frac{\partial \log p(x; \theta)}{\partial \theta_k} p(x; \theta) \, d\nu. \quad (1)$$

Rao demonstrated in his seminal paper [51] that statistical manifold can be equipped with the Fisher information metric to obtain a Riemannian structure, the study of which is known as *information geometry* [5, 4, 8]. This geometric view of statistical manifolds allows us to derive key geometric concepts for our statistical flow matching framework. For example, a *geodesic* $\gamma(t) : [0, 1] \to \mathcal{P}$ defines a "shortest" path (under the Riemannian metric) connecting two probability measures on the statistical manifold. The *geodesic distance* between two probability measures, also known as the Fisher-Rao distance [51], measures the similarity between them. The *tangent space* $T_\mu(\mathcal{P})$ at a point $\mu \in \mathcal{P}$ can be naturally identified with the affine subspace $T_\mu(\mathcal{P}) = \{\nu | \int_{\mathcal{X}} d\nu = 0\}$ where each element $\nu$ is a signed measure over $\mathcal{X}$. The *exponential map* $\exp_\mu : T_\mu(\mathcal{P}) \to \mathcal{P}$ and *logarithm map* $\log_\mu : \mathcal{P} \to T_\mu(\mathcal{P})$ can also be defined on the statistical manifold. While the geodesic for a parameterized family of probability measures can be obtained numerically by solving the geodesic equation when closed-form expression is unknown (see Appendix A.1), it usually requires expensive simulations. Fortunately, closed-form geodesic distances are available for many common distributions including categorical, multinomial, and normal distributions [44], which motivates our method.

## 2.2 Conditional Flow Matching on Riemannian Manifold

The conditional flow matching (CFM) framework [38] provides a simple yet powerful approach to generative modeling by learning a time-dependent vector field that pushes the prior noise distribution to any target data distribution. Such a flow-based model can be viewed as the continuous generalization of the score matching (diffusion) model [56, 57, 26] while allowing for a more flexible design of the denoising process. The Riemannian flow matching framework [14] further extends CFM to general manifolds on which a well-defined distance metric can be efficiently computed.

Consider a smooth Riemannian manifold $\mathcal{M}$ with the Riemannian metric $g$, a *probability path* $p_t : [0,1] \to \mathcal{P}(\mathcal{M})$ is a curve of probability densities over $\mathcal{M}$. A *flow* $\psi_t : [0,1] \times \mathcal{M} \to \mathcal{M}$ is a time-dependent diffeomorphism defined by a time-dependent vector field $u_t : [0,1] \times \mathcal{M} \to T\mathcal{M}$ via the ordinary differential equation (ODE): $\frac{\mathrm{d}}{\mathrm{d}t}\psi_t(x) = u_t(\psi_t(x))$. The flow matching objective directly regresses the vector field $u_t$ with a time-dependent neural net $v(x_t, t)$ where $x_t := \psi_t(x)$. However, this objective is generally intractable. Both [38, 14] demonstrated that a tractable objective can be derived by conditioning on the target data $x_1$ at $t = 1$ of the probability path. The Riemannian flow matching objective can be formulated as [14]

$$\mathcal{L} = \mathbb{E}_{t \sim U[0,1], x_0 \sim p_0(x), x_1 \sim q(x)}[\|v(x_t, t) - u_t(x_t|x_0, x_1)\|_g^2] \tag{2}$$

where $q$ is the data distribution, $p_0$ is the prior distribution, and $x_t := \psi_t(x|x_0, x_1)$ denotes the conditional flow. [14] further demonstrated that if the exponential map and logarithm map can be evaluated in closed-form, the condition flow can be defined as $x_t = \exp_{x_0}(t \log_{x_0} x_1)$, and the corresponding vector field can be calculated as $u_t(x_t|x_0, x_1) = \frac{\mathrm{d}}{\mathrm{d}t}x_t = \log_{x_t}(x_1)/(1-t)$. We also adapt this formulation for our statistical flow, which we will elaborate on in the next section. We note that, since we are working with manifolds of probability measures $\mathcal{M} = \mathcal{P}(\mathcal{X})$, we will use $p, q$ for probability densities *over* the manifold $\mathcal{P}(\mathcal{X})$ and $\mu, \nu$ for probability measures (or probability masses for discrete distributions) *on* the manifold $\mathcal{P}(\mathcal{X})$ to avoid confusion.

# 3 Method

Different from previous work that treated each distribution separately, we adopt an integral viewpoint toward the manifold of probability distributions. In this section, we focus on the statistical manifold of categorical distributions to demonstrate the application of our method on discrete generation tasks. However, we emphasize that our proposed SFM is applicable to any statistical manifold with a closed-form geodesic distance and can be broadly used in generative tasks targeting probability measures on the statistical manifold.

## 3.1 Statistical Manifold of Categorical Distributions

Consider the discrete sample space $\mathcal{X} = \{1, 2, \ldots, n\}$, an $n$-class categorical distribution over $\mathcal{X}$ can be parameterized by $n$ parameters $\mu_1, \mu_2, \ldots, \mu_n$ such that $\sum_{i=1}^{n} \mu_i = 1, \mu_i \geq 0$. In this way, the reference measure $\nu$ is the counting measure and the probability measure $\mu$ can be written as the convex combination of the canonical basis of Dirac measures $\{\delta^i\}_{i=1}^{n}$ over $\mathcal{X}$: $\mu = \sum_{i=1}^{n} \mu_i \delta^i$. Geometrically, this manifold can be visualized as the $(n-1)$-dimensional simplex $\Delta^{n-1}$. The geodesic distance between two categorical distributions is given in [51, 44] as

$$d_{\mathrm{cat}}(\mu, \nu) = 2 \arccos\left(\sum_{i=1}^{n} \sqrt{\mu_i \nu_i}\right). \tag{3}$$

The tangent space at a point $\mu$ can be identified with $T_\mu(\mathcal{P}) = \{u | \sum_{i=1}^{n} u_i = 0\}$ and the corresponding inner product at point $\mu$ is defined as

$$\langle u, v \rangle_\mu = \sum_{i=1}^{n} \frac{u_i v_i}{\mu_i}, \quad \mu \in \mathcal{P}_+, u, v \in T_\mu(\mathcal{P}) \tag{4}$$

where $\mathcal{P}_+$ denotes the statistical manifold of *positive* categorical distributions. Note that the inner product is ill-defined on the boundary, causing numerical issues near the boundary. To circumvent this issue, we introduce the following diffeomorphism

$$\pi : \mathcal{P} \to S_+^{n-1}, \quad \mu_i \mapsto x_i = \sqrt{\mu_i}, \tag{5}$$

which maps the original statistical manifold to the positive orthant of a unit $(n-1)$-sphere $S_+^{n-1} = \{x | \sum_{i=1}^n x_i^2 = 1, x_i \geq 0\}$ (see Fig.2). Note that we have $\|\pi(\mu)\|_2^2 = \sum_{i=1}^n \sqrt{\mu_i}^2 = \sum_{i=1}^n \mu_i = 1$. The geodesic on $S_+^{n-1}$ follows the great circle, and the following proposition holds between the geodesic distance on $S_+^{n-1}$ and $\mathcal{P}$:

**Proposition 1.**

$$d_S(\pi(\mu), \pi(\nu)) = \frac{1}{2} d_{\text{cat}}(\mu, \nu), \quad \mu, \nu \in \mathcal{P}. \tag{6}$$

A proof is provided in Appendix A.3. This indicates that we can work with the geodesic distance on the unit sphere instead with up to a constant factor:

$$d_S(x, y) = \arccos(\langle x, y \rangle), \quad x, y \in S_+^{n-1}. \tag{7}$$

The geodesic distance $d_S$ and the inner product $\langle \cdot, \cdot \rangle$ are well-defined for the boundary, and we found this transform led to the practical stabilized training of the flow model. Visualizations of the Riemannian geometry on the statistical manifold of 3-class categorical distributions and the corresponding Euclidean geometry on the simplex are provided in Fig.1 for comparison. The straight lines under the Euclidean assumptions fail to capture the true curved geometry of the statistical manifold.

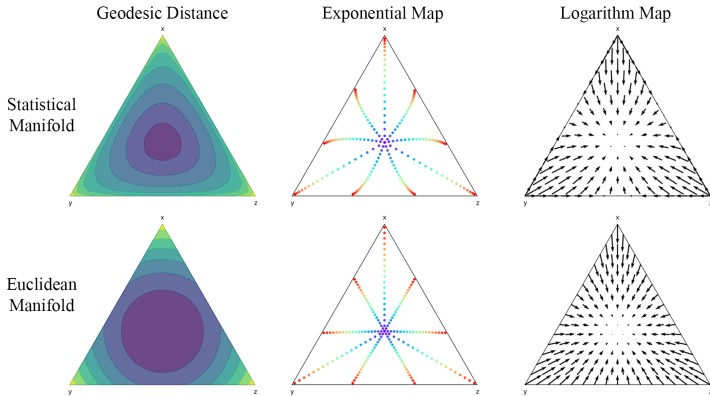

Figure 1: The Riemannian geometry of the statistical manifold for categorical distributions in comparison to Euclidean geometry on the simplex. **Left**: Contours for the geodesic distances to $\mu_0 = (1/3, 1/3, 1/3)$. **Middle**: Exponential maps (geodesics) from $\mu_0$ to different points near the boundary. **Right**: Logarithm maps (vector fields) to $\mu_0$.

### 3.2 Statistical Flow Matching

We provide the analytical form of the exponential and logarithm maps on the statistical manifold of categorical distributions in Appendix A.3. Although it is possible to directly learn the vector field following the loss in Eq.(2), such a direct parameterization has numerical issues near the boundary. As described in Sec.3.1, we apply the diffeomorphism $\pi$ in Eq.(5) to derive a more stable training objective on the spherical manifold:

$$\mathcal{L}_{\text{SFM}} = \mathbb{E}_{t \sim U[0,1], x_0 \sim \pi_*(p_0(\mu)), x_1 \sim \pi_*(q(\mu))} \left[ \|v(x_t, t) - u_t^S(x_t | x_0, x_1)\|_2^2 \right], \tag{8}$$

where $p_0$ is the prior noise distribution over $\mathcal{P}$ and $q$ is the data distribution; $\pi_*$ denotes the standard pushforward measure. $v : S_+^{n-1} \times [0, 1] \to TS_+^{n-1}$ is a learnable time-dependent vector field network that maps the interpolant $x_t$ on the unit sphere to a tangent vector in the tangent bundle $TS_+^{n-1}$. The ground truth conditional vector field $u^S$ is calculated using the exponential and logarithms maps on the sphere (Appendix A.2). The overall training and inference scheme is visualized in Fig.2 and described in Alg.2 and 3 in Appendix C.

We further implement a Euclidean flow matching model on the probability simplex with linear interpolation between the noises and the target distributions. Though linear flow matching offers "straight" lines under the Euclidean assumption, it is unaware of the intrinsic geometric properties

of the statistical manifold and turns out to trace longer paths in terms of the Riemannian metric. The objective for linear flow matching can be described as

$$\mathcal{L}_{\text{LinearFM}} = \mathbb{E}_{t \sim U[0,1], \mu_0 \sim p_0(\mu), \mu_1 \sim q(\mu)} \left[ \| v(\mu_t, t) - (\mu_1 - \mu_0) \|_2^2 \right]. \tag{9}$$

In the discussion above, we assume a single data dimension on the statistical manifold. This can be extended to any data dimension by treating them as independent channels of the input. In practice, each probability measure on the simplex is represented by a matrix $X \in [0,1]^{D \times n}$ where $D$ is the data dimension and each row sums to 1. The priors are independently sampled from the uniform distribution over the $(n-1)$-simplex and each data dimension is independently interpolated with the same timestep $t \sim U[0,1]$. The flow model, on the other hand, takes all the data dimensions as input to capture the dependence between different dimensions. During sampling, existing ODE solvers and the simple Euler method are used to integrate the vector field through time to obtain the final categorical distributions. Discrete samples are then drawn from the generated categorical distributions for evaluation. Details regarding model parameterization and sampling are further described in Appendix C.1 and C.2.

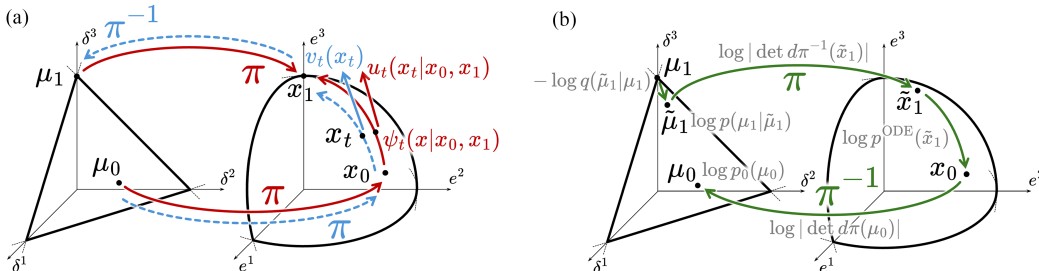

Figure 2: Statistical flow matching (SFM) framework. **(a)** During training (Sec.3.2), probability measures on $\mathcal{P}$ are mapped to $S_+^{n-1}$ via diffeomorphism $\pi$ to compute the time-dependent vector field (in red). During inference, the learned vector field generates the trajectory on $S_+^{n-1}$ and we map the outcome of ODE back to $\mathcal{P}$ (in blue). **(b)** In the NLL calculation for one-hot examples (Sec.3.5), the probability density is marginalized over a small neighborhood of some Dirac measure to avoid undefined behaviors at the boundary (in green).

## 3.3 Optimization View of Statistical Flow Matching

We further provide an interpretation of our proposed statistical flow matching framework from the perspective of an optimization problem. From the *optimization* viewpoint, for a generative model on a statistical manifold, we want to minimize the discrepancy between the target distributions and the generated distributions. Naturally, the Kullback-Leibler divergence $D_{\text{KL}}(\mu(\theta) \| \mu(\theta_1))$ can be used as a measure of statistical distance. We note that the Fisher information metric can be obtained as the Hessian of the KL divergence with respect to the parameter $\theta$. This can be demonstrated by the fact that the KL divergence reaches the global minimum of 0 at $\theta = \theta_1$, so all first-order partial derivatives are zero, and the Hessian is positive semi-definite. Therefore, Taylor expansion of KL divergence at $\theta_1$ with $\Delta\theta = \theta - \theta_1$ gives

$$
\begin{aligned}
D_{\text{KL}}(\mu(\theta) \| \mu(\theta_1)) &\approx \frac{1}{2} \sum_{jk} \Delta\theta_j \Delta\theta_k \left. \frac{\partial^2}{\partial\theta_j \partial\theta_k} D_{\text{KL}}(\mu(\theta) \| \mu(\theta_1)) \right|_{\theta=\theta_1} \\
&= \frac{1}{2} \sum_{jk} \Delta\theta_j \Delta\theta_k g_{jk}(\theta_1) = \frac{1}{2} \|\Delta\theta\|_g^2.
\end{aligned}
\tag{10}
$$

From the *geometric* viewpoint, the geodesic, by definition, is a (locally) length-minimizing curve with respect to the corresponding Riemannian metric. Therefore, by following the direction of the vector field that decreases the geodesic element $ds^2 = \| d\theta \|_g^2 \approx \|\Delta\theta\|_g^2$, we are indeed following the steepest direction that minimizes the local KL divergence. In this sense, the Fisher information metric defines a second-order optimization scheme for the KL divergence by following the "steepest" direction of the Hessian. Indeed, The steepest direction $\Delta\theta$ that decreases the KL divergence is

known as the *natural gradient* in the existing literature [2, 3] and has been explored in optimization known as *natural gradient descent*[48, 42, 46]. Instead of optimizing along the normal gradient, the natural gradient is defined as $\tilde{\nabla}\mathcal{L} = F^{-1}\nabla\mathcal{L}$ where $F$ is the Fisher information matrix estimated from a batch of sampled data. Results established in [2, 42] demonstrated that stochastic natural gradient descent is asymptotically "Fisher efficient" and is indeed the steepest descent direction in the manifold of distributions where distance is measured in small local neighborhoods by the KL divergence. Following the geodesic defined by the Fisher information metric, our SFM framework also shares these benefits with an additional advantage of analytical expressions for geodesics, as we focus on the family of categorical distributions instead of general statistical models. Such a theoretical connection may contribute to the better performance of SFM.

### 3.4 Optimal Transport on Statistical Manifold

The geometric viewpoint of the statistical manifold offers a continuous and differentiable formulation of generative modeling and also provides a robust way to measure the distance between two categorical distributions via the well-defined geodesic distance in Eq.(3). In contrast, the Markov chain-based methods cannot establish a robust distance measure due to the stochastic jumps between discrete states. Inspired by the optimal transport formulation in previous work [45, 20, 68, 63], we naturally extend it to our statistical setting in which the cost is defined by averaging the statistical distances over data dimensions as $d_{\text{cat}}(X, Y) = \frac{1}{D}\sum_{k=1}^{D} d_{\text{cat}}(\mu^{(k)}, \nu^{(k)})$ for $X = \{\mu^{(k)}\}_{k=1}^{D}, Y = \{\nu^{(k)}\}_{k=1}^{D}$. An optimal assignment of the initial noises to the target data distributions can potentially lead to more efficient training, which we demonstrated empirically in our ablation studies.

### 3.5 Exact Likelihood Calculation

Unlike diffusion-based models which rely on variational lower bounds for likelihood estimation, our proposed method shares the continuous normalizing flow's capability of exact likelihood calculation. For an arbitrary test sample $x \in \mathcal{M}$, using the change of measure formula, the likelihood can be modeled by the continuity equation [14, 43]:

$$\frac{\partial}{\partial t}\log p_t(x_t) + \text{div}_g(v_t)(x_t) = 0, \tag{11}$$

where $\text{div}_g$ is the Riemannian divergence and $v_t(x_t) := v(x_t, t)$ is the time-dependent vector field. In this way, the pushforward probability measures $p_t(x_t)$ defined via the learned flow can be obtained as the integral of the Riemannian divergence back through time on the simulated trajectory $x_t$. Following [9, 7], we define the ODE log-likelihood as the change of the log-likelihood as:

$$\log p^{\text{ODE}}(x_1) = \int_1^0 \text{div}_g(v_t)(x_t)\, dt, \tag{12}$$

where the trajectory $x_t$ is obtained by solving the differential equation $\frac{\partial}{\partial t}x_t = v_t(x_t)$ reverse through time with the initial condition $x_1$ at $t = 1$. In this way, we have $\log p(x_1) = \log p^{\text{ODE}} + \log p_0(x_0)$ where $\log p_0(x_0)$ is the log-likehood of the prior distribution at data point $x_0$. In practice, we follow previous work [7] to use Hutchinson's trace estimator [30] to efficiently obtain an unbiased estimation of the divergence using standard normal random variables. To further account for the transform $\pi$ from $\mathcal{P}$ to $S_+^{n-1}$ and the reverse transform $\pi^{-1}$, additional change of measure formulae need to be applied. Consider the pushforward of the probability measure $P$ over $\mathcal{P}$ under the diffeomorphism $\pi$ defined by $\pi_*P(V) := P(\pi^{-1}V)$, we have the change of measure identity $\pi_*P(\pi(\mu))|\det d\pi(\mu)| = P(x)$ for $x = \pi(\mu)$. Therefore, by adding the two log-determinant of the pushforward measures, the log-likelihood can be formulated as

$$\log p_1(\mu_1) = \log|\det d\pi^{-1}(x_1)| + \log p^{\text{ODE}}(x_1) + \log|\det d\pi(\mu_0)| + \log p_0(\mu_0). \tag{13}$$

The above formula is well-defined for all interior points of $\mathcal{P}$, but the change of measure terms are undefined on the boundary. This can be understood as the fact that discrete likelihoods can be arbitrarily high [62]. Following [7], we derive a variational lower bound for the likelihood as the marginalized probability over the small neighborhood of a Dirac measure $\mu_1 = \delta$ at the boundary:

$$\log p(\delta) \geq \mathbb{E}_{q(\tilde{\mu}_1|\delta)}[-\log q(\tilde{\mu}_1|\delta) + \log p(\delta|\tilde{\mu}_1) + \log p_1(\tilde{\mu}_1)], \tag{14}$$

where $q(\tilde{\mu}_1|\delta)$ can be viewed as the forward noising probability at $\tilde{\mu}_1$ which is close to $\delta$ with a closed-form likelihood. $p(\delta|\tilde{\mu}_1)$ is the categorical likelihood (cross-entropy) and $p_1(\tilde{\mu}_1)$ is the model likelihood in Eq.(13). The overall workflow of calculating NLL is demonstrated in Fig.2, and more details regarding likelihood computation can be found in Appendix B.1. It is worth mentioning that the continuous likelihood calculated here is defined with respect to the probability distribution over the statistical manifold $\mathcal{P}$. In contrast, the bits-per-dimension score for autoregressive models is usually defined with respect to a specific categorical distribution on $\mathcal{P}$ and therefore not comparable, as was also noted in [7].

## 4 Experiments

Our proposed SFM framework can be leveraged to approximate arbitrary distribution over $\mathcal{P}$, i.e., any distribution over the parameterized family of probability measures. We first demonstrate this with the toy example of a Swiss roll distribution on the probability simplex. We then conduct extensive experiments on real-world datasets for discrete generation across various domains including computer vision, natural language processing, and bioinformatics to demonstrate the effectiveness of our proposed model. We train and evaluate two different versions of the SFM framework: **SFM w/ OT** for our proposed model with optimal transport during training and **SFM w/o OT** for the model without optimal transport. A naive approach that directly works with the exponential and logarithm maps on the statistical manifold without the diffeomorphism (**SFM w/o $\pi$**) is also compared in the toy example. We also implement a linear flow matching model on the probability simplex using the loss in Eq.(9) as an additional baseline, for which we dub **LinearFM**. For discrete data, we always use Eq.(14) to obtain finite negative log-likelihood (NLL). More details regarding model architectures and the experimental setup can be found in Appendix C.

### 4.1 Toy Example: Swiss Roll on Simplex

We project the Swiss roll dataset onto the 2-simplex with some additional margins to make sure no point resides on the boundary. We used a time-dependent MLP to model the vector field for all models. The generative points on the simplex and NLL are calculated using the Dopri5 ODE solver [19] and are shown in Fig.3.

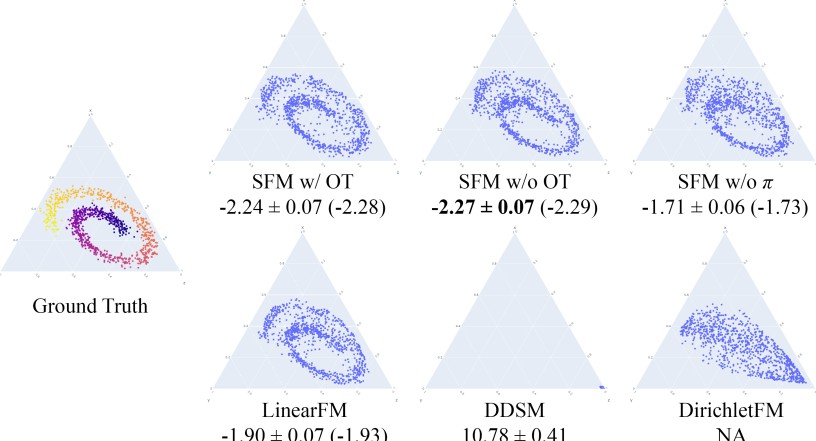

Figure 3: Generated samples of the Swiss roll on simplex dataset and NLL (lower is better). The NLLs are estimated using Hutchinson's trace estimator, whereas those in the parenthesis are exact.

We note that most existing models assume the target data to be Dirac measures (one-hot distributions) and cannot be applied to this task. Both **DDSM** [7] and **DirichletFM** [60] imposed strong prior assumptions on the data distribution as Dirichlet distributions, which failed to capture the complex geometric shape of the Swiss roll data points as demonstrated in the generated samples. On the contrary, directly built upon the geometry of the statistical manifold, our SFM model successfully captured the detailed data geometry. As no data resides on the boundary, exact NLL can be obtained using Eq.(13) and was averaged over all points in the dataset. Though linear flow matching could also capture the geometry of the data, our SFM outperformed it in terms of NLL.

## 4.2 Binarized MNIST

We extend our SFM to model discrete generation tasks in computer vision. The binarized MNIST dataset [53] is the binarized version of the original MNIST dataset [34] by thresholding the original continuous value to be either 0 or 1, thus can be viewed as a 2-class generation task with a data dimension of $28^2 = 784$. We also compared **D3PM** [6] and **DFM** [22] as examples of masked diffusion models for discrete generation. We used a convolutional neural net adopted from [57] with additional additive Fourier-based time embeddings. The quantitative evaluation of NLL and Fréchet inception distance (FID) are shown in Tab.1.

Table 1: NLL and FID of different discrete models on binarized MNIST. The NLLs in the parenthesis are discrete NLLs; therefore, they are not directly comparable. * is from [7].

| Model | SFM w/ OT | SFM w/o OT | SFM w/o $\pi$ | LinearFM |
|---|---|---|---|---|
| NLL↓ | **-1.687 ± 0.020** | -1.631 ± 0.027 | 0.928 ± 0.035 | 0.622 ± 0.022 |
| FID↓ | **4.62** | 5.15 | 8.2073 | 5.91 |
| Model | DirichletFM | DDSM | D3PM | DFM |
| NLL↓ | NA | $0.100 \pm 0.001^{*}$ | (0.141 ± 0.021) | (0.101 ± 0.017) |
| FID↓ | 77.35 | 7.79 | 67.36 | 34.42 |

All the generated results and NLL calculations were based on the Dopri5 ODE solver. Additional ablation results can be found in Appendix D.2 and generated images can be found in Appendix D.3. The NLL for diffusion-based models (D3PM and DDSM) was also calculated based on the variational bounds derived in their papers. Our proposed model consistently outperformed both the linear flow matching and other diffusion baselines in terms of FID and NLL, achieving higher sample quality and likelihood.

## 4.3 Text8

The Text8 dataset [41] is a medium-size character-level corpus consisting of a small vocabulary of 27, which includes the 26 lowercase letters and the whitespace token. We followed the convention in previous work [6, 24] to use random chunks of length 256 in both training and evaluation without any preprocessing. We used a 12-layer diffusion transformer (DiT) [49] based predictor, similar to the one used in [39]. As our exact likelihood is not directly comparable to bits-per-character (BPC) reported in previous work which relies on an alternative variational bound for the likelihood, we additionally calculate such BPC inspired by [52]. See Appendix B.2 for additional details. All generated samples and NLL were estimated using the Dopri5 ODE solver. Quantitative results of BPC are provided in Tab.2 and generated texts are provided in Appendix D.3. The results for the autoregressive language models are also provided as a reference (Discrete Flow [64] is based on autoregressive normalizing flow).

Note that, unlike all of the other baselines, SFM and LinearFM do not directly optimize such a BPC as a training objective, so such an evaluation metric is unfavorable for our model (see Appendix B.2). Despite such an unfavorable evaluation, our proposed SFM still achieved the second-best performance among other diffusion and flow baselines that were directly trained with cross-entropy losses. We also noted a significant performance gap between SFM and the naive linear flow matching on simplex, highlighting the importance of capturing the intrinsic geometric properties of the underlying statistical manifold. Optimal transport does not significantly affect the final performance here, which we presume might be due to the long training stage in which vector fields were well-explored. Additional evaluation following [12] is provided in Appendix D.1, in which SFM achieved the best generation entropy as evaluated by the pre-trained GPT-J-6B model [67].

## 4.4 Promoter Design

We further apply SFM to the practical task of promoter DNA sequence design in the bioinformatics realm. The promoter is a key element in gene transcription and regulation, and the generation of desired promoters can better help us understand the intricate interactions between human genes. [7] proposed a human promoter sequence dataset containing 100k promoter sequences with the

Table 2: BPC on Text8. Results marked [*] are taken from the corresponding papers.

| Model | BPC↓ |
|---|---|
| SFM w/ OT | $1.399 \pm 0.020$ |
| SFM w/o OT | $1.386 \pm 0.033$ |
| LinearFM | $1.651 \pm 0.027$ |
| D3PM-absorb[6] | 1.47[*] |
| BFN[24] | 1.41[*] |
| SEDD-absorb[39] | **1.32**[*] |
| MultiFlow[12] | 1.41[*] |
| Argmax Flow[28] | 1.80[*] |
| Discrete Flow[64] | 1.23[*] |
| Transformer[6] | 1.23[*] |
| Transformer XL[16] | **1.08**[*] |

Table 3: SP-MSE (as evaluated by Sei [13]) on the generated promoter DNA sequences. Results marked [*] are from [7] and results marked [†] are from [60].

| Model | SP-MSE↓ |
|---|---|
| SFM w/ OT | 0.0279 |
| SFM w/o OT | **0.0258** |
| LinearFM | 0.0282 |
| DDSM | 0.0334[*] |
| D3PM-uniform | 0.0375[*] |
| Bit-Diffusion (one-hot) [15] | 0.0395[*] |
| Bit-Diffusion (bit) [15] | 0.0414[*] |
| Language Model | 0.0333[†] |
| DirichletFM | 0.0269[†] |

corresponding transcription initiation signal profiles. Each promoter sequence is 1024 base pairs long and is centered at the annotated transcription start site position [27]. Therefore, we can model promoter design as a generation task with 4 categories (the base pair ATGC) conditioned on the given transcription signals. We follow [7] to concatenate the signal as additional input to the vector field predictor built upon 20 stacks of 1-dimensional convolutional layers on the input sequence. Optimal transport can also be applied for conditional generation, as we can pair the conditions with the input to make sure that the conditional arguments align with the target one-hot distributions.

To provide a quantitative evaluation of the generated promoter sequences, we follow [7] to apply the pre-trained deep learning sequence model Sei [13] to predict active promoters based on predictions of the chromatin mark H3K4me3. As the dataset spans the whole range of human promoter activity levels from highly expressed promoters to those with very low expression, the evaluation is based on comparing the scores between the generated sequences and the ground truth human genome promoter sequences on the test chromosomes. The metric SP-MSE is the mean squared error between the predicted promoter activity of the generated sequences and human genome sequences (lower is better) and is demonstrated in Tab.3. Our SFM was able to achieve the lowest SP-MSE score compared with other baselines. It is worth noting, though, that optimal transport produced slightly sub-optimal results. We hypothesize that this is because, in conditional generative tasks, the final generation should rely heavily on the conditions. Simply matching inputs with the targets using distance measures may discard important information in the conditional arguments and may not be the best practice.

## 5   Related Work

As we put a special interest in discrete generation, we start with the existing work on discrete generation. We first list a few traditional autoregressive models [16, 50, 64] and masked language models [18, 54] but will skip them as we mainly focus on the diffusion and flow matching models. Existing diffusion and flow-based discrete generative models can be categorized into two main groups. The first group relies on stochastic jumps of Markov chains by treating discrete classes as Markov states. By choosing a proper transition matrix, the target one-hot distribution can be gradually noised into the stationary distribution. Noticeably, this approach can also adopt an additional absorbing state to mimic the mask token in masked language modeling [6]. D3PM [6] adapted the discrete-time Markov chain with the diffusion framework and SEDD [39] proposed a more efficient training scheme by learning the score entropy between different states. [11, 55] extended it to the continuous-time Markov chain by using the infinitesimal generator (rate matrix) instead, and [12, 22] further applied the flow matching framework. Although the transition matrix or the rate matrix determines the entire diffusion trajectory, there is no canonical way of choosing it for different generation tasks to control the mixing rate. Also, due to the presence of discrete jumps of the Markov chain, exact likelihood calculation is no longer feasible. Only variational bounds were derived in [6, 12]. The other group directly works with the probability simplex or the logit space

with certain assumptions of the underlying geometric structure [36]. As an example, our linear flow matching assumes a Euclidean geometry on the probability simplex and is often used as a baseline in previous work [60]. The multinomial diffusion [28] assumed a Euclidean geometry on the logits space so interpolation became multiplication back on the probability simplex. Dirichlet diffusion (DDSM) [7] and Dirichlet flow matching (DirichletFM) [60] exerted a strong prior assumption on the probability path as the Jacobi diffusion process. We provide a more detailed analysis of these models in Appendix A.4. For models that directly operate on the logit space, the Bayesian flow network (BFN) assumed Gaussian distributions on the logit space with Bayesian update rules. [40] also assumed a Euclidean geometry in the logit space with targets as soft one-hot logits. The assumptions made in these models may not always hold, e.g., for the Swiss roll on simplex dataset in Fig.3. These assumptions also did not necessarily capture the true geometry of the underlying statistical manifold. In contrast, our proposed SFM framework does not exert any strong prior assumptions and is aware of the intrinsic geometry by following the geodesics.

We also briefly review the related work on statistical manifolds and information geometry. The field of information geometry, the study of geometric properties of statistical manifolds, was first established in Rao's seminal paper in 1945 [51] discussing the Fisher information metric. Most previous work utilizing information geometry focuses on optimization [2, 3] with a few exceptions on representation learning. [65] utilized the geodesic distance between two univariate Gaussians for function shape matching for retrosynthesis of gold nanoparticles. [35] treated point cloud data as probability measures over the 3D Euclidean space and considered the pullback metric on the latent space to obtain optimal latent coordinates for the autoencoder. [47] further applied such a method on single-cell RNA sequence trajectories. These models demonstrated the advantage of following the proper geometry compared with the vanilla Euclidean setting, but the pullback metric needed to be evaluated with expensive Jacobian-vector products during training. In contrast, our proposed SFM, for the first time, leverages mathematical tools from information geometry to generative modeling. SFM directly operates on the statistical manifold with closed-form geodesics, providing a simulation-free approach for efficient generative training.

## 6 Discussion

In this paper, we proposed statistical flow matching (SFM) as a general generative framework for generative modeling on the statistical manifold of probability measures. By leveraging results from information geometry, our SFM effectively captures the underlying intrinsic geometric properties of the statistical manifold. Specifically, we focused on the statistical manifold of categorical distributions in this work and derived optimal transport and exact likelihood formulae. We applied SFM to diverse downstream discrete generation tasks across different domains to demonstrate our framework's effectiveness over the baselines. We noted that SFM can be further extended to non-discrete generative tasks whose targets are probability distributions, which we will leave as future work.

We are also aware of the limitations of our SFM framework. As a special case of the flow matching model, the generation is an iterative process of refinement that cannot modify the size of the initial input. This may pose limitations to generation compared with autoregressive models. Furthermore, we adopted the assumption of independence between classes so that the canonical Riemannian structure can be induced by the Fisher metric. However, discretized data like CIFAR-10 [32] (256 ordinal pixel values) do not follow this assumption, and results on such data might be suboptimal.

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

# Supplementary Material

## A  Riemannian Manifold and Information Geometry

In this section, we give a more detailed mathematical background on Riemannian manifold and information geometry related to this work. A more comprehensive background on the Riemannian manifold can be found in [21]. For information geometry, [8, 4] provide more comprehensive analyses and rigorous formulations.

### A.1  Riemannian Manifold

A Riemannian manifold $\mathcal{M}$ is a real, smooth manifold equipped with a positive definite inner product $g$ on the tangent space $T_x(\mathcal{M})$ at each point $x \in \mathcal{M}$. Let $T\mathcal{M} = \bigcup_{x \in \mathcal{M}} T_x(\mathcal{M})$ be the *tangent bundle* of the manifold $\mathcal{M}$, a time-dependent *vector field* on $\mathcal{M}$ is a mapping $u_t : [0,1] \times \mathcal{M} \to T\mathcal{M}$ where $u_t(x) \in T_x(\mathcal{M})$. A *geodesic* is a locally distance-minimizing curve on the manifold. The existence and the uniqueness of the geodesic state that for any point $x \in \mathcal{M}$ and for any tangent vector $u \in T_x(\mathcal{M})$, there exists a unique geodesic $\gamma : [0,1] \to \mathcal{M}$ such that $\gamma(0) = x$ and $\gamma'(0) = u$. The *exponential map* $\exp : \mathcal{M} \times T\mathcal{M} \to \mathcal{M}$ is uniquely defined to be $\exp_x(u) := \gamma(1)$. The *logarithm map* $\log : \mathcal{M} \times \mathcal{M} \to T\mathcal{M}$ is defined as the inverse mapping of the exponential map such that $\exp_x(\log_x(y)) \equiv y, \forall x, y \in \mathcal{M}$. Note that the formulation here is slightly different from Sec.2.1 where we fix a point $\mu$ on $\mathcal{P}$. With the exponential map and logarithm map, the time-dependent flow can be compactly written as time interpolation along the geodesic:

$$x_t := \psi_t(x_t | x_0, x_1) = \exp_{x_0}(t \log_{x_0} x_1), \quad t \in [0,1]. \tag{15}$$

It can be demonstrated that the above flow indeed traces the geodesic between $x_0, x_1$ with linearly decreasing geodesic distance $d_g(x_t, x_1) = (1-t) d_g(x_0, x_1)$ [14]. For an $n$-dimensional Riemannian manifold, the geodesic can be obtained by solving the geodesic equation (using the Einstein summation convention):

$$\frac{\mathrm{d}^2 x^k}{\mathrm{d}t^2} + \Gamma_{ij}^k \frac{\mathrm{d}x^i}{\mathrm{d}t} \frac{\mathrm{d}x^j}{\mathrm{d}t} = 0, \quad k = 1, 2, \ldots, n, \tag{16}$$

where $x^i$ are local coordinates of the geodesic and $\Gamma_{ij}^k$ are the Christoffel symbols defined by

$$\Gamma_{ij}^k = \frac{1}{2} g^{km} \left( \frac{\partial g_{mi}}{\partial x^j} + \frac{\partial g_{mj}}{\partial x^i} - \frac{\partial g_{ij}}{\partial x^m} \right), \quad i, j, k = 1, 2, \ldots, n, \tag{17}$$

where $g^{ij}$ is the inverse metric such that $g^{ij} g_{jk} = \delta_k^i$. The closed-form expressions for the geodesic are generally not available. [44] curated a wide range of common statistical manifolds for parameterized families of both the discrete and the continuous distributions. We will focus on the statistical manifold of categorical distributions and also the spherical manifold, as the latter is closely related to the former via the diffeomorphism $\pi$ in Eq.(5).

### A.2  Spherical Manifold

The positive orthant of the unit $(n-1)$-sphere $S_+^{n-1}$ is a $(n-1)$-dimensional manifold. The sphere can be embedded into the ambient Euclidean manifold $\mathbb{R}^n$ such that it inherits the canonical inner product as

$$\langle u, v \rangle_S = \langle u, v \rangle = \sum_{i=1}^n u_i v_i, \quad u, v \in T_x(S_+^{n-1}). \tag{18}$$

The tangent space $T_x(S_+^{n-1}) = \{u | \langle u, x \rangle = 0\}$ is a $(n-1)$-dimensional hyperplane perpendicular to the vector $x$. The geodesic on the sphere follows the great circle between two points, and the geodesic distance can be calculated in Eq.(7). The corresponding exponential map can be calculated as:

$$\exp_x(u) = x \cos \|u\|_2 + u \operatorname{sinc} \|u\|_2, \tag{19}$$

where $\operatorname{sinc}(\theta) = \sin \theta / \theta$ is the unnormalized sinc function. The logarithm map can be calculated as:

$$\log_x(y) = \arccos(\langle x, y \rangle) \frac{P_x(y - x)}{\|P_x(y - x)\|_2}, \tag{20}$$

where $P_x(w) = w - \langle x, w \rangle x$ is the projection of vector $w$ onto the tangent space $T_x(S_+^{n-1})$.

### A.3 Statistical Manifold of Categorical Distributions

By calculating the Fisher information matrix for a categorical distribution, we can obtain the explicit form of the metric tensor as [44]

$$g_{jk}(\mu) = \frac{\delta_{jk}}{\mu_j} + \frac{1}{\mu_n}, \quad 1 \le i, j, \le n - 1, \tag{21}$$

where $\delta_{jk}$ is the Kronecker delta. Substituting this into $\langle u, v \rangle_\mu$ leads to the Riemannian inner product in Eq.(4). The statistical manifold of categorical distributions $\mathcal{P}(\mathcal{X})$ is closely related to $S_+^{n-1}$ by the diffeomorphism $\pi$ in Eq.(5). In fact, $\pi$ is an isometry between the Fisher information metric on $\mathcal{P}$ and the standard induced metric on $S_+^{n-1}$ up to a constant scaling factor of $1/4$. To see this, we have the following proposition:

**Proposition 2.**

$$\left\langle \frac{\partial \pi(\mu)}{\partial u}, \frac{\partial \pi(\mu)}{\partial v} \right\rangle = \frac{1}{4} \langle u, v \rangle_\mu, \quad u, v \in T_\mu(\mathcal{P}). \tag{22}$$

*Proof.*

$$\left\langle \frac{\partial \pi(\mu)}{\partial u}, \frac{\partial \pi(\mu)}{\partial v} \right\rangle = \left\langle \frac{\mathrm{d}}{\mathrm{d}t} \pi(\mu + tu) \Big|_{t=0}, \frac{\mathrm{d}}{\mathrm{d}t} \pi(\mu + tv) \Big|_{t=0} \right\rangle = \sum_{i=1}^n \frac{u_i}{2\sqrt{\mu_i}} \frac{v_i}{2\sqrt{\mu_i}} = \frac{1}{4} \langle u, v \rangle_\mu.$$

$\square$

Note that the infinitesimal arc length on the geodesic can be expressed as $\mathrm{d}s^2 = \|\mathrm{d}x\|_g^2 = \sum_{jk} \mathrm{d}x_j g_{jk}(x) \mathrm{d}x_k$ where $\|\cdot\|_g$ is the canonical Riemannian norm induced by the inner product. Integrating over the geodesic, we can easily arrive at the result in Proposition 1. Based on this, the exponential map and logarithm map on this statistical manifold can be written as

$$\exp_\mu(u) = \left( \sqrt{\mu} \cos \left\| \frac{u}{2\sqrt{\mu}} \right\|_2 + \frac{u}{2\sqrt{\mu}} \operatorname{sinc} \left\| \frac{u}{2\sqrt{\mu}} \right\|_2 \right)^2, \tag{23}$$

$$\log_\mu(\nu) = \frac{2 \arccos(\langle \sqrt{\mu}, \sqrt{\nu} \rangle)}{\sqrt{1 - \langle \sqrt{\mu}, \sqrt{\nu} \rangle}} (\sqrt{\mu \odot \nu} - \langle \sqrt{\mu}, \sqrt{\nu} \rangle \mu), \tag{24}$$

where $\odot$ denotes the pairwise multiplication. We mentioned in the main text that directly using the exponential and logarithm maps on the statistical manifold has numerical issues near the boundary. Instead, we used the mapping $\pi$ to work with the spherical manifold. Nonetheless, Eq.(23) and (24) provide useful tools for visualization of the statistical manifold, as demonstrated in Fig.1.

We specifically note the property that the Riemannian structure induced by the Fisher information metric leads to vector fields more parallel to the boundaries. This can also be derived mathematically from the logarithm map in Eq.(24). Consider the direction term $\sqrt{\mu \odot \nu} - \langle \sqrt{\mu}, \sqrt{\nu} \rangle \mu$. For $\mu$ close to the boundary with $\mu_k \approx 0$, its corresponding vector field will also have a close to $u_k \approx 0$ component, which is different from linear flow matching's fixed $\nu - \mu$. We hypothesize that one potential benefit of such a curved geometry over the naive Euclidean geometry is that the former helps prevent overshooting across the boundaries. Specifically, consider a target point on the boundary. The Euclidean vector field will continue to push the points outside the manifold, whereas the Riemannian vector field tends to travel parallel to that boundary to prevent going across the boundary. Once overshooting happens during sampling, the model may exhibit undefined behavior as it was never trained on the points outside the manifold.

We also noted the relation between the Fisher information metric defined in Eq.(1) and the canonical inner product defined in Eq.(4) for general statistical manifolds. Let $\mathcal{M}$ be an $n$-dimensional differentiable manifold and consider an embedding defined by $\mathcal{M} \hookrightarrow \mathcal{P}(\mathcal{M}), \theta \mapsto p(\theta) = \sum_{i \in \mathcal{X}} p_i(\theta) \delta^i$.

The pullback of the Fisher metric $g$ defines a metric on $\mathcal{M}$. For $u, v \in T_\theta(\mathcal{M})$, we have

$$
\begin{aligned}
g_\theta(u, v) &:= p^*(g)_\theta(u, v) \\
&= g_{p(\theta)}\left(\frac{\partial p}{\partial u}, \frac{\partial p}{\partial v}\right) \\
&= \sum_i \frac{1}{p_i(\theta)} \frac{\partial p_i}{\partial u}(\theta) \frac{\partial p_i}{\partial v}(\theta) \\
&= \sum_i p_i(\theta) \frac{\partial \log p_i}{\partial u}(\theta) \frac{\partial \log p_i}{\partial v}(\theta)
\end{aligned}
\tag{25}
$$

which gives the more common definition of the Fisher information matrix in Eq.(1). This relation also generally holds for a continuous sample space $\mathcal{X}$ but requires more careful handling with measure theory tools. We refer interested readers to mathematical references, e.g., Chapter 3.1 in [8].

### A.4 Other Statistical Manifolds

The statistical manifold of categorical distributions provides an intuitive way of modeling discrete targets. However, we noted that other parameterized distributions can also be used to derive alternative statistical manifolds with different geometries. We will briefly discuss some manifolds that have been used in previous generative models (though not from the information geometry perspective as we do).

**Dirichlet Distribution**  The Dirichlet distribution $\mathrm{Dir}(\alpha)$ is a multivariate generalization of the beta distribution. It is defined on the $(n-1)$-dimensional simplex $\Delta^{n-1}$ with $n$ concentration parameters $\alpha = \{\alpha_i\}_{i=1}^n$. The probability density function is defined as

$$
p(x; \alpha) = \frac{1}{\mathrm{B}(\alpha)} \prod_{i=1}^n x_i^{\alpha_i - 1}, \quad x \in \Delta^{n-1}
\tag{26}
$$

where $\mathrm{B}(\alpha) = \prod_{i=1}^n \Gamma(\alpha_i)/\Gamma(\sum_{i=1}^n \alpha_i)$ is the multivariate beta function. The closed-form expressions for the geodesics between Dirichlet distributions (and for the marginal beta distributions, as were used in [7]) are unknown. It is known that, however, this statistical manifold has non-constant negative sectional curvatures everywhere [33]. [7] proposed to follow the specific path of Jacobi diffusion processes on the marginal beta distributions, for which expensive modified Jacobi polynomials needed to be precomputed. [60] further applied the flow matching framework by considering the probability path of $\mathrm{Dir}(1 + te_k)$ for some fixed maximum timestep. Still, expensive calculations of the regularized incomplete beta functions were required. Compared with these two generative models, our proposed SFM on categorical distributions has the following advantages:

- DDSM and DirichletFM have strong prior assumptions that the categorical distributions follow some specific forward noising process which always terminates at (or near) one-hot distributions. This assumption generally holds for discrete generation, but cannot be generalized to more complex geometry on the probability simplex, as we have demonstrated in our toy example of the Swiss roll on simplex dataset in Fig.3.

- The forward noising paths are not the shortest geodesics in the sense of the Riemannian metric. In our SFM, the vector fields always follow the geodesic, thus following the sharpest direction of decreasing KL divergence (see Sec.3.3).

- Our SFM framework does not require the computation of expensive polynomials and is more mathematically concise.

- The final one-hot distributions are unreachable in [60], as it would require an infinite timestep. Therefore, a variational bound needs to be derived for likelihood estimation, which was not done in the original paper.

**Multinomial Distribution**  Consider $m$ i.i.d. experiments that follow a categorical distribution with $n$ possible outcomes and probability $\mu = \{\mu_i\}_{i=1}^n$, a multinomial distribution gives the proba-

bility of getting $x_i$ times the $i$-th outcome with $\sum_{i=1}^{n} x_i = m$. The geodesic distance for multinomial distributions is identical to that of the categorical distributions up to a scaling constant [44]:

$$d_{\mathrm{mul}}(\mu, \nu) = \sqrt{n} d_{\mathrm{cat}}(\mu, \nu) = 2\sqrt{n} \arccos\left( \sum_{i=1}^{n} \sqrt{\mu_i \nu_i} \right). \tag{27}$$

Therefore, our model can also be interpreted as multinomial flow matching on the corresponding statistical manifold. The previous work of multinomial diffusion (argmax flow) [28] transformed the one-hot distribution to $ne_k - 1$ in the logit space (soft one-hot logits) and assumed the common Euclidean structure of the logit space for constructing the diffusion process. This assumption did not consider the intrinsic geometry of the underlying statistical manifold and also had the inaccessibility issue of the Dirac measure as DirichletFM. We have demonstrated that SFM consistently outperformed the naive multinomial diffusion on the Text8 datasets, indicating the effectiveness of tracing the true geodesics.

# B  Likelihood Calculation

In this section, we provide more concrete mathematical details regarding the exact likelihood calculation in the main text. We first make a distinction between negative log-likelihood (NLL) and bits-per-dimension (BPD, including bits-per-character, BPC) in this work. As we are dealing with probability distributions over the statistical manifold of probability measures, it is important for us to keep the difference between the probability spaces $\mathcal{P}(\mathcal{P}(\mathcal{X}))$ and $\mathcal{P}(\mathcal{X})$. In our notation, we use $\mu, \nu$ to denote elements in $\mathcal{P}(\mathcal{X})$, or categorical distributions, and use $p, q$ to denote elements in $\mathcal{P}(\mathcal{P}(\mathcal{X}))$, or distributions over categorical distributions. We further use NLL to refer to $-\log p(\mu)$ for a specific $\mu$ and BPD for categorical likelihood $-\log_2 p(\delta|\mu)$ where $\delta$ is the ground truth Dirac measure (one-hot distribution). Note this definition of BPD (BPC) follows the convention in autoregressive language models where the conditional probabilities for the next token are calculated and averaged [16, 23]. In contrast, most previous work drew equivalence between these two concepts.

## B.1  Negative Log-Likelihood (NLL) Calculation

Our SFM framework enjoys the exact negative log-likelihood calculation as a continuous normalizing flow model [38, 14]. It is worth noting that negative NLLs are not uncommon [14]. As a probability density, the NLL can indeed go to negative infinity at the boundary. Consider the arcsine distribution defined on the finite support $[0, 1]$ with the probability density function

$$p(\theta) = \frac{1}{\pi\sqrt{\theta(1-\theta)}}, \quad \theta \in (0, 1). \tag{28}$$

It is easy to see the density approaches infinity as $\theta$ approaches 0 or 1. Note that the arcsine distribution is a special case of beta distribution $\mathrm{Beta}(\frac{1}{2}, \frac{1}{2})$, which is in turn a special case of Dirichlet distribution $\mathrm{Dir}(\frac{1}{2}, \frac{1}{2})$. Each $\theta$ can be naturally viewed as the parameter for the Bernoulli distribution (2-class categorical distribution), so the arcsine distribution indeed defines a distribution over Bernoulli distributions with negative NLLs near the boundary. In fact, for discrete data, the target data always come in as Dirac measures, which indicates that the target distribution must be a convex combination of Dirac measures with infinite likelihood: $q = \sum_{i=1}^{n} \theta_i \delta_i$ where $\delta_i := \delta_{\delta^i}$ is the Dirac measure over the underlying Dirac measure $\delta^i$ on the discrete class $i$.

**Variational Bound**  We followed [7] to derive a variational bound as marginalized over a small neighborhood of the Dirac measure in Eq.(14) to obtain finite NLLs. The variational bound can be

derived via the standard variational approach as

$$
\begin{aligned}
\log p(\delta) &= \mathbb{E}_{q(\mu|\delta)}\left[\log p(\delta)\right] \\
&= \mathbb{E}_{q(\mu|\delta)}\left[\log\left(\frac{p(\delta,\mu)}{p(\mu|\delta)}\right)\right] \\
&= \mathbb{E}_{q(\mu|\delta)}\left[\log\left(\frac{p(\delta,\mu)}{q(\mu|\delta)}\frac{q(\delta,\mu)}{p(\mu|\delta)}\right)\right] \\
&= \mathbb{E}_{q(\mu|\delta)}\left[\log\left(\frac{p(\delta,\mu)}{q(\mu|\delta)}\right)\right] + D_{\mathrm{KL}}(q(\mu|\delta)\|p(\mu|\delta)) \\
&\geq \mathbb{E}_{q(\mu|\delta)}\left[\log\left(\frac{p(\delta,\mu)}{q(\mu|\delta)}\right)\right] \\
&= \mathbb{E}_{q(\mu|\delta)}\left[\log p(\delta|\mu) + \log p(\mu) - \log q(\mu|\delta)\right].
\end{aligned}
\tag{29}
$$

The posterior probability $q(\mu|\delta)$ serves as the forward diffusion process in [6, 60] with a closed-form expression that depends on the timestep. In our Riemannian flow matching formulation, however, such a forward process is implicitly defined via the time-interpolation along the geodesics. Nonetheless, we do know that the conditional distribution should approach the Dirac measure when $t \to 1$ as the geodesic distance approaches 0: $p_1(\mu|\delta) \approx \delta$. Therefore, as we are marginalized over a small neighborhood, we can define a time-dependent posterior distribution $q_t(\mu|\delta)$ that approaches $\delta$ as $t \to 1$. In principle, any forward probability that approaches the Dirac measure at $t = 1$ with an analytical likelihood can be used to derive the lower bound. For example, we can follow [7] to use the Jacobi diffusion process. To avoid the need for expensive Jacobi polynomial calculation, we instead use simple indicator measures over the simplex as

$$
q_t(\mu|\delta^k) = \begin{cases} \frac{p_0}{(1-t)^{n-1}}, & \mu_i \leq t, 1 \leq i \leq n, i \neq k \\ 0, & \text{otherwise} \end{cases}
\tag{30}
$$

where $p_0$ is the base probability of the uniform distribution over the simplex, $\mu_i = \theta_i$ is the $i$-th component of the categorical distribution of $\mu$. In other words, $q_t(\mu|\delta^k)$ is the time-interpolated probability between the Dirac measure and the uniform distribution: $q_t(\mu|\delta^k) = t\delta_k + (1-t)U\Delta^{n-1}$. We have

$$
\log q_t(\mu|\delta^k) = \log p_0 - (n-1)\log(1-t).
\tag{31}
$$

In practice, we use $t = 0.995$ and provide more results in the ablation studies in Appendix D.2. The categorical likelihood $p(\delta^k|\mu)$ is the standard cross-entropy loss, also referred to as the *reconstruction loss* in some previous work [24]. It can be calculated as

$$
\log p(\delta^k|\mu) = \log\mu_k.
\tag{32}
$$

**Prior Likelihood**  The prior likelihood accounts for the base probability when we sample from the prior noise distribution. During both the training and the inference stages, we uniformly sample from the simplex $\Delta^{n-1}$. This can be efficiently done by normalizing $n$ i.i.d random variables from the exponential distribution:

$$
\tilde{\theta}_i \sim \mathrm{Exp}(1), \theta_i = \tilde{\theta}_i / \sum_{k=1}^{n}\tilde{\theta}_k, \quad i = 1, 2, \ldots, n.
\tag{33}
$$

The uniform distribution over simplex is exactly the Dirichlet distribution $\mathrm{Dir}(1,\ldots,1)$. Therefore, the log-likelihood can be calculated as

$$
\log p_0 = \log\Gamma(n)
\tag{34}
$$

where $\Gamma$ is the Gamma function. Note that the prior likelihood is independent of the samples.

**Change of Measure**  The diffeomorphism $\pi$ in Eq.(5) induces a pushforward measure on $S_+^{n-1}$ which can be described by the change of measure identity

$$
\pi_* P(\pi(\mu)) |\det \mathrm{d}\pi(\mu)| = P(x), x = \pi(\mu).
\tag{35}
$$

The change of measure term can be calculated using the change of volume formula with the canonical coordinates $\{\tilde{x}_i\}_{i=1}^{n-1}$ and $\{\varphi_i\}_{i=1}^{n-1}$ as

$$
\begin{aligned}
\mu_1 &= \tilde{x}_1, & x_1 &= \cos\varphi_1 \\
\mu_2 &= \tilde{x}_2, & x_2 &= \sin\varphi_1\cos\varphi_2 \\
&\vdots & &\vdots \\
\mu_{n-1} &= \tilde{x}_{n-1}, & x_{n-1} &= \sin\varphi_1\ldots\sin\varphi_{n-2}\cos\varphi_{n-1}
\end{aligned}
\tag{36}
$$

where $0 \le \varphi_i \le \pi/2$. The diffeomorphism $x = \pi(\mu)$ is given by

$$
\begin{aligned}
\tilde{x}_1 &= \cos^2\varphi_1 \\
\tilde{x}_2 &= \sin^2\varphi_1\cos^2\varphi_2 \\
&\vdots \\
\tilde{x}_{n-1} &= \sin^2\varphi_1\ldots\sin^2\varphi_{n-2}\cos^2\varphi_{n-1}.
\end{aligned}
\tag{37}
$$

The change of measure term is

$$
\log|\det d\pi^{-1}(x)| = -(n-1)\log 2 - \sum_{i=1}^{n}\log x_i.
\tag{38}
$$

Similarly, for the inverse mapping, the term is

$$
\log|\det d\pi(\mu)| = (n-1)\log 2 + \frac{1}{2}\sum_{i=1}^{n}\log\mu_i.
\tag{39}
$$

**Model Likelihood**   The model likelihood can be viewed as the pushforward measure of the learned flow:

$$
p_t = (\psi_t)_* p_0.
\tag{40}
$$

We demonstrated in Sec.3.5 that this term can be calculated as the integral of divergence in Eq.(12) plus the base probability in Eq.(34). More specifically, the NLL can be obtained as the solution to the following ODE system back through time with the initial condition of $x_1$ and $\log p_1^{\text{ODE}} = 0$:

$$
\frac{d}{dt}\begin{bmatrix} x_t \\ \log p_t^{\text{ODE}} \end{bmatrix} = \begin{bmatrix} v_t(x_t) \\ -\text{div}_g(v_t)(x_t) \end{bmatrix}.
\tag{41}
$$

For manifolds that can be embedded in the ambient Euclidean space (e.g., simplex and sphere), the Riemannian divergence can be calculated as the normal Euclidean divergence $\text{div}_g = \text{div}$ [14]. We further use Hutchinson's trace estimator [30] to efficiently estimate the divergence as

$$
\text{div}\, v_t(x) = \mathbb{E}_{\varepsilon\sim\mathcal{N}(0,I)}[\varepsilon^\top \nabla v_t(x)\varepsilon].
\tag{42}
$$

The expectation can be efficiently computed using the vector-Jacobian product. For data with a unitary dimension $D = 1$ (e.g., Swiss roll on simplex), we also calculate the exact divergence using the equality $\text{div}\, v = \text{Tr}(J(v))$, where $J(v)$ is the Jacobian of the vector field $v$. The computational cost for the exact divergence calculation scales quadratically with the data dimension $D$, as the interdependence between data dimensions makes it computationally expensive to calculate the Jacobian. Therefore, we only demonstrated the NLL with exact divergence calculation for the Swiss roll dataset in Fig.3, in which Hutchinson's trace estimator could make a decent estimation of the exact divergence.

Combining Eq.(31), (32), (34), (38), and (39) with Eq.(14) and (13), we can efficiently calculate the NLL given arbitrary Dirac measures as input. The NLL calculation scheme is visualized in Fig.2 and described in Alg.1.

## B.2   Bits-Per-Character (BPC) Calculation

We have demonstrated that the NLL can reach negative infinity. In contrast, the common definition of BPC with the cross-entropy loss $-\log_2 p(\delta|\mu)$ is always non-negative and consistent with previous

**Algorithm 1** NLL Calculation for Discrete Data
___

1: Sample $\tilde{\mu}_1 \sim q_t(\mu|\delta)$ in Eq.(30) and calculate $-\log q_t(\tilde{\mu}_1|\delta)$ and $\log p(\delta|\tilde{\mu}_1)$.
2: Apply the diffeomorphism in Eq.(5) to obtain $\tilde{x}_1 = \pi(\tilde{\mu}_1)$ and calculate $\log|\det \mathrm{d}\pi^{-1}(\tilde{x}_1)|$.
3: Solve the ODE system in Eq.(41) to obtain $x_0$ and $\log p^{\mathrm{ODE}}$.
4: Apply $\pi^{-1}$ to obtain $\mu_0 = \pi^{-1}(x_0)$ and calculate $\log|\det \mathrm{d}\pi(\mu_0)|$.
5: Calculate the base log probability $\log p_0(\mu_0)$.
6: **return** NLL as in Eq.(14).
___

autoregressive language models [62]. However, as our flow-based model is not autoregressive, we instead follow the variational formulation in [52] to calculate the alternative NLL and BPC for comparison. The ELBO in [52] can be calculated as

$$\mathcal{L}_{\mathrm{ELBO}} = -\mathbb{E}_{\mu_1 \sim q(\mu)}\left[\int_0^1 \frac{1}{1-t}\log\langle\hat{\mu}_1(\mu_t),\mu_1\rangle\,\mathrm{d}t\right] \tag{43}$$

where $\mu_t = \exp_{\mu_0}(t\log_{\mu_0}\mu_1), \mu_0 \sim p_0(\mu)$ denotes the interpolation along the geodesic and $\hat{\mu}_1(\mu_t)$ denotes the learned model's prediction for $t = 1$ given the current noise data $\mu_t$. This can be efficiently done by one-step prediction with the learned vector field $v$ as

$$\hat{\mu}_1(\mu_t) = \exp_{\mu_t}\left((1-t)v_t(\mu_t)\right). \tag{44}$$

The standard Euclidean inner product is used in Eq.(43) so it can be understood as a weighted cross-entropy loss. Note that the integral in Eq.(43) has a singularity at $t = 1$, making it numerically unstable to estimate using ODE solvers. Instead, we follow [52] to apply the change of variable $s = -\log(1-t)$ to reformulate the ELBO as

$$\mathcal{L}_{\mathrm{ELBO}} = -\mathbb{E}_{\mu_1 \sim q(\mu)}\left[\int_0^\infty \log\langle\hat{\mu}_1(\mu_t),\mu_1\rangle\,\mathrm{d}s\right] \tag{45}$$

where $t = 1 - e^{-s}$. Note that for large $s$, the corresponding $t$ is very close to 1, so the integrand is very close to zero. Indeed, $s = 10$ corresponds to $1 - t < 5 \times 10^{-5}$, so we simply set the upper limit to 10 and used the Dopri5 solver to numerically estimate the ELBO. BPC can be then calculated as $\mathcal{L}_{\mathrm{ELBO}}/\log 2$. This formulation also shares a similar form as the other ELBOs derived in [6, 12], and is thus comparable to most existing models.

Eq.(43) was directly optimized in Markov chain-based methods like D3PM [6], MultiFlow [12], and SEDD [39], and also autoregressive language models. In contrast, our SFM (and LinearFM) follows the continuous normalizing flow setup in [38], which enables the exact likelihood calculation instead of ELBOs. Therefore, trained on the alternative objective of minimizing the Riemannian vector field norm in Eq.(8), SFM did not directly try to minimize such an ELBO. Despite such an unfavorable evaluation compared to other baselines, SFM was still able to achieve the second-best BPC, as demonstrated in Tab.2.

### B.3 GPT-J-6B Likelihood

GPT-J-6B [67] is a transformer-based large language model with 6B trainable parameters. We follow the pipeline in [12] to first tokenize the generated text using the provided tokenizer with GPT-J-6B. The GPT-J-6B NLL is then calculated based on the predicted logits on the token level using the pre-trained model as:

$$\mathcal{L}_{\mathrm{GPT}} = -\frac{1}{K}\sum_{k=1}^K \log p\left(w_k|w_{1:k-1}\right), \tag{46}$$

where $w$ are tokens and $K$ is the number of tokens. Similarly, the entropy is calculated on the empirical distribution of the tokens based on a large number of generated texts. We noted that, as GPT-J-6B was not pre-trained on Text8, its NLL does not necessarily reflect the true data distribution in Text8, as we will demonstrate more concretely in the evaluation plot in Appendix D.1 and generated samples in Appendix D.3.

## C  Experimental Setup

In this section, we further describe the experimental setup, model architecture, and datasets.

## C.1 Model Parameterization

Our SFM architecture is encoder-agnostic and can be applied to arbitrary discrete generative tasks. Here, we describe the common setting of the flow model across different datasets. We use the sinusoidal timestep embedding described in [66] as $\text{Emb} : [0, 1] \to \mathbb{R}^H$. We follow [14] to manually project the predicted vector field onto the corresponding tangent space. For the spherical manifold, the projection can be described as

$$v_t(x_t) = \tilde{v}_t(x_t) - \langle x_t, \tilde{v}_t(x_t) \rangle x_t. \tag{47}$$

For linear flow matching on the simplex, the projection can be described as

$$v_t(x_t) = \tilde{v}_t(x_t) - \frac{1}{n} \sum_{i=1}^n \tilde{v}_t(x_t)_i. \tag{48}$$

The projection guarantees that the final prediction lies on the corresponding tangent space, and the data points will stay on the manifold during Euler sampling. We will always assume the projection is performed in the following context of using $v_t$. The training stage of SFM is described in Alg.2. The time complexity of our SFM framework is dominated by the underlying vector field predictor with little overhead. Each model for binarized MNIST and promoter design was trained on a single 80GB NVIDIA A100 GPU for 6-10 hours. Each model for Text8 was trained on four 80GB NVIDIA A100 GPUs for about 7 days. We will further describe the encoders for each generation task in the following subsections.

---

**Algorithm 2** Training SFM

---

1: **while** not converged **do**
2:     Sample noise distribution $\mu_0 \sim p_0(\mu)$ and target distribution $\mu_1 \sim q(\mu)$.
3:     **if** optimal transport **then**
4:         Do batch OT assignments of $\mu_0$ and $\mu_1$ according to the average statistical distances.
5:     **end if**
6:     Apply the diffeomorphism in Eq.(5) to obtain $x_0 = \pi(\mu_0), x_1 = \pi(\mu_1)$.
7:     Sample $t \sim U[0, 1]$ and interpolate $x_t = \exp_{x_0}(t \log_{x_0} x_1)$ using Eq.(19) and (20).
8:     Calculate the conditional vector field $u_t^S(x_t | x_0, x_1) = \frac{\mathrm{d}}{\mathrm{d}t} x_t = \log_{x_t}(x_1)/(1 - t)$.
9:     Predict the vector field using $v(x_t, t)$ and optimize the SFM loss in Eq.(8).
10: **end while**

---

## C.2 Model Sampling

The sampling process from the trained model can be described as solving the differential equation $\frac{\partial}{\partial t} x_t = v_t(x_t)$ from $t = 0$ to $1$ with the initial conditional $x_0$ sampled from the prior noise distribution. Alternatively, we can write the solution as the integral of the learned time-dependent vector through time as

$$x_1 = x_0 + \int_0^1 v_t(x_t) \, \mathrm{d}t. \tag{49}$$

We always use the Dopri5 ODE solver [19] for our ODE sampling and NLL calculation. For Euler method with $N$ discrete steps, the timesteps $0, 1/N, 2/N, \ldots, (N-1)/N$ are used with a step size of $1/N$. The sampling stage is described in Alg.3.

## C.3 Swiss Roll

The Swiss roll on the 2-simplex is generated by normalizing the span of the 2-dimensional Swiss roll to $[0 + \varepsilon, 1 - \varepsilon]$ where $\varepsilon$ is a small margin to make sure no point lies on the boundary. The third dimensional is automatically obtained as $\mu_3 = 1 - \mu_1 - \mu_2$. We fixed a random seed and generated 1000 samples for a full batch training. The vector field predictor is based on simple multi-layer perceptions (MLPs) and can be described as

$$v(x_t, t) = \text{MLP}(\text{MLP}(x_t) \| \text{MLP}(\text{Emb}(t))) \tag{50}$$

where $\|$ denotes concatenation and we used a hidden dimension of $H = 128$. Each model was trained for 2000 epochs using full batch gradient descent with an initial learning rate of $10^{-3}$. 1000

---

**Algorithm 3** Sampling from SFM

---

1: Sample noise distribution $\mu_0 \sim p_0(\mu)$.
2: Apply the diffeomorphism in Eq.(5) to obtain $x_0 = \pi(\mu_0)$.
3: **if** ODE sampling **then**
4:     Solve $\frac{\partial}{\partial t} x_t = v_t(x_t)$ using Dopri5 ODE solver with initial condition $x_0$.
5: **else**                                                                 ▷ Euler method
6:     **for** $t \leftarrow 0, 1/N, 2/N, \ldots, (N-1)/N$ **do**
7:         $x_{t+1/N} = \exp_{x_t}(v(x_t, t)/N)$
8:     **end for**
9: **end if**
10: **return** $\mu_1 = \pi^{-1}(x_1)$

---

samples were sampled using the Dopri5 ODE solver, and NLL (based on Hutchinson's trace estimator) was calculated on the whole training data with 20 repeats. The DirichletFM model was originally trained with cross-entropy loss which could not be used for this task. We instead used the binary cross-entropy loss during training and kept all the other flow-based part the same as described in the original paper [60].

## C.4 Binarized MNIST

We used the preprocessed binarized MNIST dataset from [53] which has a split of 50k/10k/10k. We adopted the CNN-based vector field predictor from [57] with additional additive time embeddings at each convolution layer's output. The model has 4 residual layers [25] and 4 refinement layers [37] with a total number of 29.8M trainable parameters. All models were trained for 100k iterations with a batch size of 256 (approximately 510 epochs) with an initial learning rate of $3 \times 10^{-4}$.

For the quantitative evaluation of the generated samples, we calculated the FID score for 1000 generated samples for each model. The statistics for the binarized MNIST dataset were calculated on the whole training set via the pre-trained InceptionV3 [61] model. The NLLs reported in the main text were calculated on a fixed subset with 1000 samples of the test set using the Dopri5 ODE solver. See Appendix D.3 for generated images.

## C.5 Text8

We followed previous work [24, 6] to use a fixed split of 90M/5M/5M with a fixed sequence length of 256. We used a 12-layer diffusion transformer (DiT) [49] based predictor, similar to the one used in [39]. The resultant model has 92.4M trainable parameters. Noticeably, our model size is smaller than [24] which used a full 24-layer model, but is similar to [39]. The models were trained for a total number of 3M iterations with a batch size of 512 per GPU (approximately 16 epochs), an initial learning rate of $10^{-4}$, and an exponential moving average (EMA) decay rate of 0.9999. We further used 1000 iterations as the linear warmup of the learning rate and used a decay rate of 0.8 with a patience of 3 and a validation interval of 2000 training iterations. The model snapshot with the lowest validation loss was saved for evaluation. BPCs were calculated on the first 4k sequences of length 256 of the test set (about 1M characters) with the Dopri5 solver as described in Appendix B.2, and generated texts were also obtained using the Dopri5 ODE solver. GPT-J-6B NLL and the generation token entropy were calculated based on 4k generations of length 256 (about 1M characters). See Appendix D.3 for generated texts.

## C.6 Promoter Design

We used the splits from the dataset paper [7] that assign Chromosome 10 to the validation set, Chromosomes 8 and 9 to the test set, and all the other 21 human chromosomes to the training set. This assignment can avoid potential data leakage on the same chromosome. The transcription signals are provided as a float number for each base pair position. We also followed [7] to use a total number of 100k sequences with a context window of 1024 base pairs and added a random offset of 10 to the sequence position during training. The vector field predictor we used was identical to that in [7], with 20 stacks of 1D convolutional layers and a total number of 13.3M trainable parameters. Our models were trained for 200k iterations with a batch size of 256 and an initial learning rate

of $5 \times 10^{-4}$. The model snapshot with the lowest validation SP-MSE on the validation set was saved for evaluation. The test SP-MSE was evaluated on the generated samples on the full test set's transcription signals with 300 Euler steps of generation.

# D    Additional Result

In this section, we provide additional results of the evaluation on Text8, additional ablation studies, and generated samples for each task.

## D.1    Additional Result on Text8

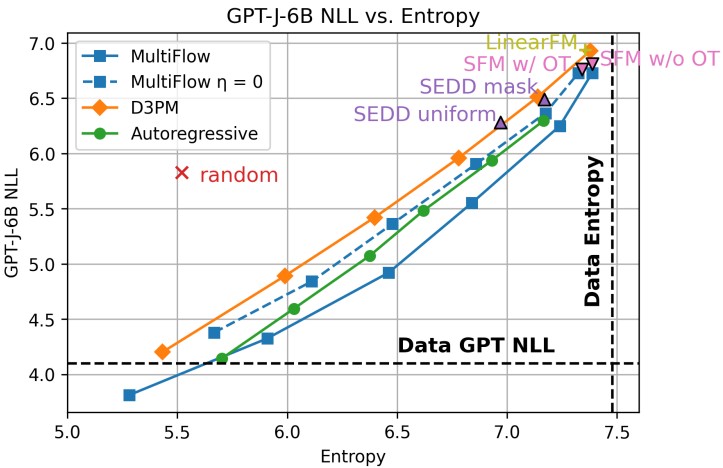

Figure 4: GPT-J-6B NLL versus sample entropy. For MultiFlow, D3PM, and autoregressive, the curve represents different logit temperatures from 0.5 to 1. Baseline data are from [12].

We further follow [12] (MultiFlow) to calculate the NLL using GPT-J-6B [67], a pre-trained autoregressive large language model. As such an NLL can be easily fooled by repeating high-frequency words, we also follow [12] to report additional token entropy, for which a closer entropy to data is preferred. The GPT-J-6B NLL and the token entropy are plotted in Fig.4, with the raw baseline data and the permission to reproduce provided by the authors of [12]. An additional random baseline that generates each character independently based on the letter frequency was provided. The ground truth data NLL and entropy are also included as the dotted horizontal and vertical lines. They demonstrate the best results any model can possibly achieve. Our SFM tended to better capture the diversity of the text corpus with the best entropy. Generated texts are provided in Appendix D.3.

As GPT-J-6B was not trained on Text8, we noted a caveat that its NLL may not necessarily reflect the Text8 distribution fitness. From the MultiFlow results, such an NLL can be made artificially low by duplicating high-frequency words, e.g., repeated numbers with little semantic meaning as in the low-temperature MultiFlow generations (see Appendix D.3 for examples). We also noted that such an NLL can be easily fooled by randomly generated strings based on letter frequency. Additionally, low-temperature MultiFlow variants achieved lower NLLs than the ground truth data, making this metric less credible. We noted the huge impact of temperature on MultiFlow as it generated more repeated numbers with lower temperatures. In contrast, SFM achieved perceptually similar or even better results with more diversity.

## D.2    Ablation Study

We provide ablation studies of the effect of different sampling methods, different sampling steps for the Euler methods, and different $t_{\max} \approx 1$ for NLL calculation. The results are provided in Tab.4. The effect of the variational timestep $t_{\max}$ matched the results from [7], which stated that a closer timestep to the target data would decrease the NLL. The reasons we choose $t = 0.995$ instead of higher values are: 1) we noticed numerical stability issues at a timestep very close to the target, 2) we want a fair comparison with [7] which used an equivalent timestep of 0.9975, and 3) as long as

we use a same timestep, the results are comparable within. We also found that NLL increased with the number of Euler sampling steps. We hypothesize that it is due to the large divergence change at $t \approx 1$ and naive Euler steps tend to overestimate its contribution. Nonetheless, the Euler method with more than 300 steps gave a similar NLL as the ODE solver. We further provide additional results of the NLLs and FID scores on different sample methods in Tab.5 in which we used $t = 0.995$ and 300 Euler steps. The final performance was quite similar between these two sampling methods.

Table 4: NLL for different sampling methods, sampling steps, and $t_{\max}$.

| Method | #step | $t_{\max}$ | NLL↓ |
|---|---|---|---|
| Euler | 300 | 0.9995 | -2.545 ± 0.029 |
| | | 0.999 | -2.442 ± 0.018 |
| | | 0.995 | -1.689 ± 0.031 |
| | | 0.99 | -1.019 ± 0.032 |
| | 100 | 0.995 | -1.871 ± 0.020 |
| | 500 | | -1.677 ± 0.022 |
| | 1000 | | -1.647 ± 0.024 |
| ODE | - | 0.995 | -1.687 ± 0.020 |

Table 5: NLL and FID for different sampling methods with $t_{\max} = 0.995$. For the Euler method, we used 300 Euler steps.

| Method | Model | NLL↓ | FID↓ |
|---|---|---|---|
| ODE | SFM w/ OT | -1.687 ± 0.020 | 4.62 |
| | SFM w/o OT | -1.631 ± 0.027 | 5.15 |
| | LinearFM | 0.622 ± 0.022 | 5.91 |
| Euler | SFM w/ OT | -1.689 ± 0.031 | 5.00 |
| | SFM w/o OT | -1.653 ± 0.028 | 4.86 |
| | LinearFM | 0.499 ± 0.022 | 6.47 |

## D.3 Generated Samples

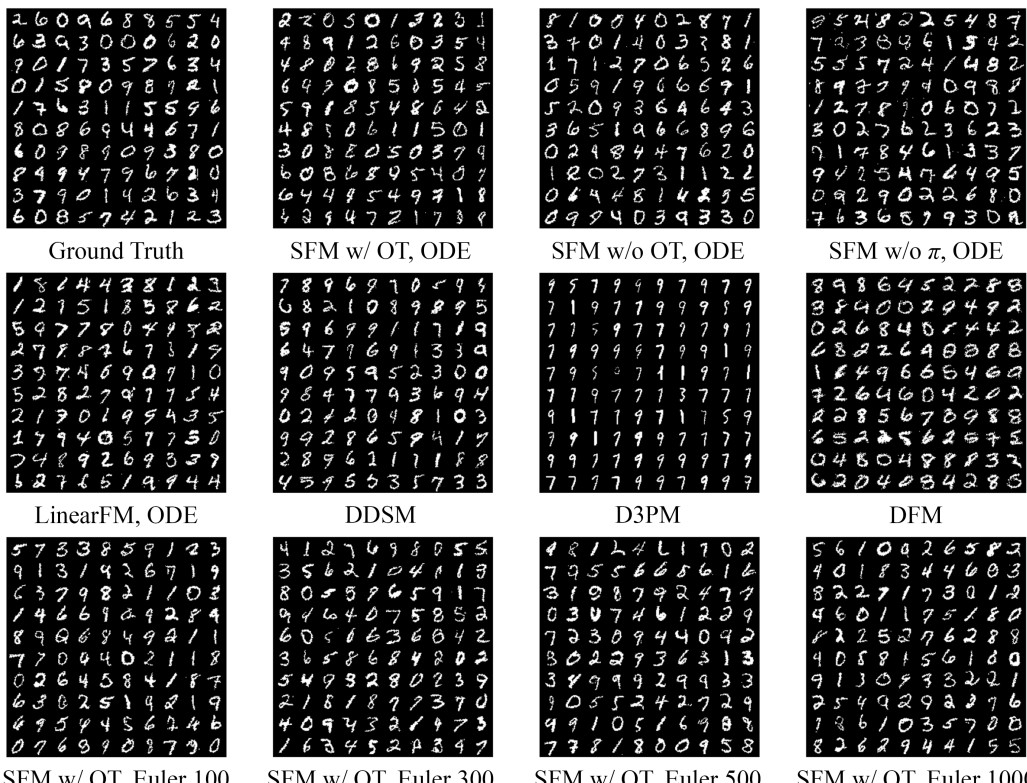

Figure 5: Generated samples of the binarized MNIST dataset from various models and different sampling settings.

The ground truth images and generated samples of the binarized MNIST dataset using various models and different sampling settings are demonstrated in Fig.5. The lower row demonstrates generated samples with different numbers of Euler steps.

For the Text8 dataset, generated texts, NLL, entropy, and per sample NLL evaluated by GPT-J-6B using different models and sampling methods are demonstrated in Tab.6, with one sample with an NLL chosen around the average, one above the average, and one below the average for each model to provide a more comprehensive demonstration of the generation. The MultiFlow with different sampling temperatures used the model checkpoint in [12]. As SFM directly samples categorical probabilities instead of making multiple discrete jumps with logits, it is incompatible with temperature-tuning.

Table 6: Generated samples and GPT-J-6B NLLs. For each model, the average NLL and entropy calculated on 4k generations are also provided. The NLL marked for each generation is the sample NLL, as evaluated by GPT-J-6B.

SFM w/ OT, ODE, NLL: 6.762, Entropy: 7.340

| | |
|---|---|
| zero_zero_zero_more_as_well_as_the_needed_of_all_of_it_church_the_country_s_higner_upcoming_bank_the_country_comment_on_quebec_e dits_includes_the_account_of_diego_hyle_ciaspare_coes_tain_three_zero_seven_zero_millimeter_if_south_of_the_south_leo_jordan_the | NLL: 6.336 |
| such_as_in_outcarge_of_coincination_with_mows_such_as_adler_martie_the_hilly_patt_evedhon_of_morcele_s_night_of_blood_the_tremen t_of_eliensberg_while_an_ulav_at_esrheim_that_he_had_to_proved_left_mainied_this_label_is_in_hellenistic_separatism_the_falix_ro | NLL: 6.805 |
| r_t_orator_lemmoi_s_mother_toury_ghost_for_his_history_on_a_blaster_the_three_stallman_family_sources_including_the_film_that_a_ro mance_nine_author_higtly_lacaded_the_second_harmour_open_source_for_which_orrie_changed_the_bluebogs_books_moy_s_athlite_s_medit | NLL: 7.522 |

SFM w/o OT, ODE, NLL: 6.811, Entropy: 7.387

| | |
|---|---|
| _became_known_as_the_shacon_valley_to_the_heaven_green_and_in_the_middle_of_the_lechneit_tracked_the_line_kej_nis_a_valley_one_p inochules_this_was_verified_by_many_charterly_brollary_applications_including_those_which_synonymous_with_orbits_some_of_the_mas | NLL: 6.407 |
| cable_now_masi_had_little_to_port_from_six_eight_nine_made_hofavor_a_new_printer_of_disruption_this_platforv_would_be_faving_to_ the_current_country_but_this_need_for_saw_della_even_this_four_one_three_bit_moil_callers_did_soo_after_a_as_n_if_platform_for_t | NLL: 6.819 |
| nomic_ancestor_wh_meil_berg_hiarst_red_rthonstrak_utter_upon_technology_baddendin_models_on_bendrays_hypothesies_anti_aer_dynami cs_work_have_been_intelligent_to_develop_an_european_astronomic_conifice_in_the_production_of_ten_conifices_of_develop_and_princ | NLL: 7.479 |

LinearFM, ODE, NLL: 6.935, Entropy: 7.356

| | |
|---|---|
| is_resulted_in_gawzik_college_in_the_five_season_of_feason_at_twice_the_atmosphere_is_named_after_the_list_called_him_before_inn _s_college_at_stulpford_university_of_london_also_cambridge_the_burroughs_henrians_college_which_is_yelled_apollo_one_college_na | NLL: 6.466 |
| ne_two_eight_zero_perhaps_that_one_s_stream_roman_frxwuapered_the_practices_of_telleeist_speakership_settled_and_an_army_of_the_ two_set_of_love_relationships_the_foundation_of_the_colfederation_homewater_to_during_the_civil_war_or_dan_brown_xian_john_zinso | NLL: 6.935 |
| level_mortans_already_sick_but_evade_dissolve_the_moses_of_auctional_with_deng_about_four_sekes_there_was_a_moikade_problem_to_p eople_who_receive_signed_grief_of_culture_of_the_middle_bone_island_for_a_more_designation_of_a_kick_trade_bands_and_rangers_bom | NLL: 7.454 |

MultiFlow, $T = 1$, NLL: 6.728, Entropy: 7.387

| | |
|---|---|
| er_of_the_soap_opera_by_andrew_wills_goosecat_productions_one_nine_nine_one_the_sea_monsters_of_the_late_one_nine_nine_zero_s_th e_famous_woman_stanley_goodman_jerdre_mcnabb_out_of_zoom_movie_barry_leroy_barbara_lewis_and_brenda_punceco_aka_sney_steary_aka_ | NLL: 5.906 |
| she_hill_obhalarnnach_eochrans_eileann_munthar_cearlomha_mhaonna__tardare_mho_mord_tore_lian_linn_mu_phaile_gael_cangallauig_lao thuis_guilleith_leois_glion_guildh_lara_gall_innonte_tilbonne_guilecht_shuachtansert_guillaste_guatnaoic_asthache_cuichant_conai | NLL: 6.648 |
| lde_its_replacement_and_or_not_mist_mere_decabeod_man_and_drast_m_ek_or_ubangostrades_dialogue_or_connon_cainne_as_follows_make_ wolsey_conane_i_get_clean_to_contemplate_the_static_problem_to_reduce_it_into_perception_for_frbellist_man_jewish_views_the_othe | NLL: 7.426 |

MultiFlow, $T = 0.9$, NLL: 6.249, Entropy: 7.240

| | |
|---|---|
| inism_weaver_routledge_new_york_w_g_toland_one_nine_nine_two_women_texts_gender_history_raymond_lynn_lucky_henry_pecher_one_nine _eight_eight_the_modern_women_lyceum_new_york_harper_mead_one_nine_seven_eight_wharke_george_one_nine_nine_one_modern_history_ex | NLL: 5.277 |
| eight_it_deplets_the_size_of_the_starters_of_the_high_land_of_the_new_tahpu_co_bon_monte_tucals_and_quitla_land_as_elen_de_las_a tlas_as_landous_pierce_the_torch_crack_into_steamy_places_to_eatons_hence_to_weed_out_their_blood_from_them_and_urge_them_to_hea | NLL: 6.352 |
| ion_of_jack_molice_now_fise_bob_springfield_slurs_boiling_funny_fruit_feed_and_chasing_rocked_duos_off_r_e_s_life_both_sides_had _an_unwanted_and_violent_violence_and_the_rage_gained_a_recent_show_the_trial_was_the_clash_of_john_d_d_the_first_trial_at_which | NLL: 7.074 |

MultiFlow, $T = 0.8$, NLL: 5.552, Entropy: 6.840

| | |
|---|---|
| ht_six_four_alexander_leroda_russian_writer_d_one_nine_one_nine_one_eight_six_seven_daniel_chase_american_author_d_one_nine_thre e_two_one_eight_seven_eight_adolf_luroda_austrian_composer_d_one_nine_six_four_one_eight_eight_two_robert_w_keisler_american_pub | NLL: 4.090 |
| umadhushah_ashara_uladineshah_sahiraj_singh_one_five_seven_nine_rajnav_singh_ajmandharajaghar_singh_rohanjit_singh_one_five_eigh t_zero_sardoyar_teachings_of_narshenhara_singh_one_five_eight_one_pranajmahr_jaharkara_jogyar_grandson_of_ujearjeer_one_five_nin | NLL: 5.223 |
| y_daughter_god_originates_to_come_as_a_pot_of_inspiration_all_one_of_whom_shall_witness_the_extortion_unto_you_and_will_ask_your _neighbour_do_you_hear_and_rempskyour_lord_and_all_our_love_are_wise_tough_hope_speaked_and_thy_beautiful_names_rather_cool_when | NLL: 6.840 |

