# OpenReview forum: "Categorical Flow Matching on Statistical Manifolds"
_NeurIPS.cc/2024/Conference — NeurIPS 2024 poster_

### Official Review · Reviewer_WPWv · 2024-07-09

**Soundness:** 3
**Presentation:** 3
**Contribution:** 3
**Rating:** 6
**Confidence:** 5

**Summary:**

The paper extends Flow Matching to discrete ("categorical") variables, similarly to a whole flourish of other papers that appeared in the last months, all of which work on the "probability simplex", i.e. they pass from a discrete alphabet A to the finite dimensional probability space P(A), which can be identified with a simplex with vertex set A. Here the emphasis is on the Fisher-Rao metric over this simplex, which the authors claim gives a natural riemannian metric that is well-suited to the flow-matching formalism.

**Strengths:**

The method is well founded and the geometric insights are inspiring and seem "right".

It seems that the method performs well, as indicated by experiments.

The paper is well written and clear, easy to read execept for a few hiccups (mentioned in "questions" section).

**Weaknesses:**

The Riemannian metric on the subset of the sphere $\{x\in \mathbb R^n: |x|^2=2, x_i\geq 0 \text{ for }1\le i\le n\}$ is not that different (in fact a $C^1$-close deformation of) the one on the actual "flat" simplex. So why would the results be so much different between the two cases? The grounding for a big gap from this setup to the standard one is not clear.

It is also not clear why this method would have to perform better than Stark et al. "Dirichlet flow" paper.

**Questions:**

By the way, what's the difference between this paper and https://arxiv.org/abs/2405.14664 ?

I understand and follow well the core paper, so I have only a few minor questions:
1) line 94: what does it mean to "condition on the delta measure"? A measure is not a random variable right? I don't fully follow what this means.
2) line 151: "we assume a single data dimension" -- what does that mean? also line 152 "extended to any data dimension" is kind of confusing, but I'm sure this is minor.. please clarify?
3) for section 3.3, line 171 and following, says "the Fisher information metric defines a second-order optimization [...]".. I don't fully understand what this means exactly. A metric is a metric, it doesn't have agency and so it doesn't define anything. I think this could be clarified by adding a few more formulas and by expanding on what the authors actually mean.
4) also for section 3.4 I don't fully follow: what Optimal Transport are the authors considering, how is that set up and how is that justified? At the moment I don't find this section very useful to a reader.
5) about section 3.5: why would one want to calculate NLL? The reason for that is missing, so I think it would make sense to shorten the derivation of the NLL formula to a minimum (move them to the appendix?) and actually explain where this section is coming from, as it's not crucial to actual flow matching objectives, and at the moment seems a bit artificial tbh

**Limitations:**

see "weaknesses" section.

---

> ### Author Rebuttal · Authors · 2024-08-07
>
> We thank you for your high recognition of our work's novel geometric insight and clear delivery. We will address your questions and concerns as follows.
>
> ## Q1 Advantage of Riemannian structure
>
> Unlike LinearFM which assumes a flat simplex and straight flows, SFM considers the Riemannian structure induced by the Fisher information metric for categorical distributions. The **naive Euclidean assumption may fail to capture the true geometry** of the statistical manifold. Furthermore, we demonstrated in Sec.3.3 that **following the vector field induced by the Fisher information metric coincides with the steepest direction of the natural gradient that decreases the KL divergence**, which may also contribute to a better performance from the optimization point of view. Another possible advantage comes from the curved geodesic in Fig.1. Unlike straight geodesic under the Euclidean setting, the Riemannian vector fields are curved towards the boundary (more parallel to the boundary), making it hard to *overshoot* across the boundary.
>
>
> ## Q2 Comparison to DirichletFM
>
> We have included comparisons to DirichletFM in Appendix A.4 to provide some insights into SFM's better performance. DirichletFM considered the specific probability path in which **the target distributions are always assumed to be Dirichlet**. Such an assumption is not applicable to more complex target distributions and fails the Swiss roll toy example in Fig.3. Additionally, its probability path does not reflect the shortest geodesic distance on the statistical manifold, which may lead to suboptimal flows and vector fields. In contrast, our SFM framework constructs flows **based on the geodesic on the statistical manifold**, where vector fields follow the steepest direction of decreasing KL divergence. Our method can applied to any source and target distribution, making it more flexible and efficient.
>
>
> ## Q3 FisherFlow
>
> We thank you for mentioning FisherFlow, which is a concurrent work closely related to ours. We note that the FisherFlow paper was uploaded on May 23 after the NeurIPS submission ddl. Therefore, **SFM and FisherFlow should be considered concurrent work**.
>
> Both SFM and FisherFlow explored the Riemannian structure of the statistical manifold to establish flows and vector fields by constructing isomorphic mappings between the simplex and the sphere. We both demonstrated the theoretical connection to natural gradient and explored the minibatch-OT formulation during training. **We took a step further to derive the exact likelihood over the probability space and a tight ELBO for the likelihood of discrete data**. We carefully choose the prior, reference measure, and divergence to ensure comparability with other discrete diffusion/flow models.
>
> While FisherFlow predominantly focused on bioinformatic tasks of DNA design, **we conducted more comprehensive experiments across various domains with additional baselines** to explore the superior performance and versatility of our method. We believe both works are important explorations on the use of Riemannian geometry structure on discrete generative tasks.
>
>
> ## Q4 Conditional on measure
>
> We apologize for the typo. We meaned "condition on $x_1$" where the conditional probability path has $p_1(x|x_1)\approx \delta_{x_1}$ at $t=1$, as described in the flow matching paper. In our case, each target point $x_1$ is a one-hot distribution $\mu_1$.
>
>
> ## Q5 data dimension
>
> The data dimension refers to the length of data. As an example, a DNA sequence with $D$ bases has data dimension $D$. We can represent this data as a matrix $X \in [0,1]^{D \times n}$, where $n=4$ reflects the four different categories (A, T, C, G). For simplicity, we assumed $D=1$ in the derivations, which is a standard practice. It is straightforward to extend this to multi-dimension by jointly modeling $D$ categorical probability measures.
>
> ## Q6 Connection to natural gradient
>
> Please see the common rebuttal.
>
>
> ## Q7 Optimal transport
>
> We consider minibatch-OT similar to [20,43,62] but on the statistical manifold. We match between a minibatch of the noises from the prior distribution and the samples from the target distribution with the smallest transport cost (defined line 189) based on the statistical distance defined in Eq.3. A thorough investigation of the theoretical benefit of minibatch OT can be found in the original paper. For Markov-chain based models like D3PM and MultiFlow, it is not possible to derive such a distance measure due to the discrete jump between Markov states.
>
> ## Q8 Significance of NLL calculation
>
> Generative models are trained to capture data distribution $p_\theta\approx p_\text{data}$. Therefore, it is natural and crucial to calculate the likelihood for a given data sample $p_\theta(x)$ as both **the evaluation metric for generative models and confidence quantification for data**. When evaluated on ground truth data, the NLL serves as a natural evaluation metric of how closely the generative model fits the data. Many common evaluation metrics including cross-entropy, perplexity, and bits-per-dimension are derived from likelihood estimation. With the ability to calculate likelihood, we can provide an intrinsic evaluation of flow models.
>
> Additionally, NLL can be used to measure the confidence of given inputs and facilitate RLHF of flow models. Many policy-based RL requires explicit log-likelihood, and ELBOs that are loose (e.g. with impromptu noise schedule) could impact the effectiveness of RLHF. Our exact NLL formulation is more accurate and our ELBO definition over discrete data is arguably much tighter, potentially benefiting applications that explicitly rely on likelihood.

---

> ### Comment · Reviewer_WPWv · 2024-08-10
>
> I thank the authors for the rebuttal, but I am not understanding one of the answers.
>
> About the sphere metric, I don't see how taking a simplex and curving the interior (mapping from a flat simplex to a curved one) can prevent overshooting over the boundary. Could you please elaborate?
>
> To clarify my doubt, think of the case of 3 categories. We then compare a flat equilateral triangle with straight lines, to a "quadrant" of a sphere, where geodesics on a sphere are so to say, "pieces of equators", or maximal circunferences on the sphere. Then the segment going from a vertex of the straight triangle to the opposite side, makes an angle $\alpha\in[\pi/3, \pi/2]$ with the boundary whereas, in the curved setting, the geodesic from a vertex of the quadrant to the opposite side makes an angle $\alpha=\pi/2$ with the boundary. So why does one trajectory overshoot less than the other?
>
> I can see the performance of your experiments, but to me the geometry of the sphere quadrant and of a simplex still seem quite similar. In fact, adding to what I said in the review: the two metric spaces are not just $C^1$-diffeomorphic, but actually have finite distortion one with respect to the other, and I think the distortion constant is smaller than $2$. So, what would warrant such a big difference of performance then?
>
> Given the above comparison of agles of geodesics, I don't think the answer is related to overshooting, it must be something else.
>
> Emphasizing again, to be clear: this is just an important curiosity for me, because I can see the better performance of the experiments. But still I'm not convinced about the actual principle/reason beyond the better performance at the moment.

---

> > ### Author Response · Authors · 2024-08-10
> >
> > We thank your feedback for our rebuttal and we are more than happy to further discuss the potential benefits of considering the curved geometry induced by the Fisher information metric. We will further elaborate on our hypothesis of overshooting for SFM versus LinearFM.
> >
> > In the middle column of Fig.1, we plotted the geodesics between the same source and target pairs under the Euclidean setting (flat simplex) and the Riemannian settings (using the Fisher information metric). Besides the curved geodesics, we also noted that the spacing between adjacent points is also different (though they are linearly spaced with respect to the geodesic distance). It can be seen from the figure that points are **more clustered near the boundary than those near the middle region**. Also note that the ground truth conditional vector fields have constant length for all timestep $t$ for both the flat simplex (always $\mu_1-\mu_0$) and the sphere (the arc length between $x_0,x_1$). In this way, a predicted vector field on the sphere with the same norm near the boundary will instead move the point for a smaller distance on the simplex to avoid overshooting. The Euclidean vector field, on the other hand, remains constant near the boundary.
> >
> > Furthermore, for points near the boundary but not at the vertex (which may occur in our toy example), it can be seen from Fig.1 that the vector fields and the geodesic are **curved to be more parallel to the boundaries**. It can be also demonstrated mathematically (in our response to Reviewer xYE5) by looking at the direction term $\sqrt{\mu\odot\nu}-\langle\sqrt{\mu},\sqrt{\nu}\rangle \mu$. For $\mu$ close to the boundary with component $\mu_k\approx 0$, its corresponding vector field will also have a close to $u_k\approx 0$ component, which is different from linear flow matching's fixed $\nu-\mu$. In this way. the Riemannian vector field avoids further pushing the points outside the boundaries.
> >
> > We further noted that the **geodesic distance in Eq.3 cannot be bounded by the Euclidean distance**. More rigorously, there does not exist a finite constant $C$ such that $d\_\text{cat}(\mu,\nu)\le C\\|\mu-\nu\\|\_2,\forall \mu, \nu$. In other words, there isn't a finite distortion constant due to the singularity of the transform on the boundary. This can be demonstrated with Taylor expansion at $\mu$ as:
> > $$
> > d_\text{cat}(\mu,\mu+\Delta\mu)=2\arccos\left(\sum_{i=1}^n\sqrt{\mu_i(\mu_i+\Delta\mu_i)}\right)
> > \approx 2\arccos\left(\sum_{i=1}^n\mu_i+\frac{\Delta\mu_i}{2}-\frac{\Delta\mu_i^2}{8\mu_i}\right)
> > =2\arccos\left(1-\sum_{i=1}^n\frac{\Delta\mu_i^2}{8\mu_i}\right)
> > \approx2\sqrt{2}\sqrt{\sum_{i=1}^n\frac{\Delta\mu_i^2}{8\mu_i}}=\sqrt{\sum_{i=1}^n\frac{\Delta\mu_i^2}{\mu_i}}
> > $$
> > Compared to the Euclidean distance $\\|\Delta\mu\\|=\sqrt{\sum_{i=1}^n\Delta\mu_i^2}$, it is clear that $d_\text{cat}(\mu,\mu+\Delta\mu)$ cannot be bounded when some $\mu_i$ is close to zero. An alternative theoretical benefit of SFM is its connection to natural gradient, as we have mentioned in the common rebuttal.
> >
> > We also noted that similar non-flat assumptions can be found in previous work including DirichletFM which follows the specific path of Dirichlet distributions and achieves better performance than LinearFM. Again, we sincerely thank you for bringing in such an inspirational discussion and we hope our explanation addresses your questions.

---

> > > ### Comment · Reviewer_WPWv · 2024-08-10
> > >
> > > Sorry but I'm not convinced yet about the overshooting justification. Do you maybe have some empirical evidence about this?
> > >
> > > About Fig. 1, I have the suspicion that it deforms the sphere quadrant nontrivially, in order to plot it to a simplex, and that deformation does not preserve angles and gives the wrong visual impression. Precisely: for geodesics starting from the center, in the case of the simplex, the angles geodesics form with the boundary are in the range $[\pi/6, \pi/2]$ and for the sphere quadrant they are in the range $[\pi/4, \pi/2]$. Angles in the second case are higher than in the first case, so geodesics are less parallel to the boundary for the sphere, going against the "overshooting becomes better" argument. Correct me if I got the intervals wrong. They come from the following observation: angles of a "spherical triangle" are of $\pi/2$ so when a geodesic goes from the center to a vertex (which is the geodesic making the smallest angle with the boundary), it makes an angle of half of $\pi/2$, namely $\pi/4$. The same computation for the "flat simplex" (an equilateral triangle) gives angle of $\pi/6$.
> > >
> > > Another point, maybe this helps for some clarification: I don't follow your computation about "sphere quadrant distance blowing up", I think it doesn't blow up.
> > >
> > > $2\mathrm{arccos}(\langle x,y\rangle)$ (formula (7) in the paper) is twice the angle between vectors $x,y\in \mathbb S^{d-1}_2$ (sphere of radius $2$) so this says that you are actually working with a sphere with usual geodesic distance (this distance is radius=2, times angle).
> > >
> > > On the other hand, you wrote $\mathrm{arccos}(\sqrt{\langle x,y\rangle})$ in your above computation. Is there a mismatch? Of course for small angle $\epsilon>0$ we have "distance $\approx$ radius times angle", but if you put the square root, you get the quantity looking like $\sqrt{\epsilon}/ \epsilon$, and that becomes arbitrarily high as $\epsilon\to 0$, but I don't see where the square root comes from.
> > >
> > > I'm still convinced that you work with a statistical manifold isometric to sphere quadrant, that the curvature of the sphere slightly helps. I think that the distortion between that and a simplex is controlled, a factor between 1 and 2, which still gives a nontrivial increase in performance. Can this be accurate?
> > >
> > > I'm happy to discuss, if you have time to discuss more on this. It's not a big issue with the paper, but just want to get to the bottom of it. Please give higher priority to other rebuttals if you want.

---

> > > > ### Author Response · Authors · 2024-08-11
> > > >
> > > > We thank your prompt reply and we are happy to continue this inspirational discussion with you. We first want to assure you that Fig.1 was **accurately plotted using the exponential and logarithm maps of the Riemannian simplex equipped with Fisher-Rao metric directly** (Eq.22 & 23 in Appendix A.3) with geodesic $\mu_t=\exp_{\mu_0}(t\log_{\mu_0}\mu_1)$ (should be equivalent to Eq.3). In other words, this visualization is not made from deforming a sphere quadrant, although such derivation should result in the same outcome. We note that the Riemannian structure can be defined directly on the simplex, we can call it *Riemannian simplex* (equipped with the Fisher information metric, and geodesic as Eq.3), which is different from a *flat simplex* (equipped with the Euclidean metric, as in LinearFM). The diffeomorphism to the unit sphere was introduced later in Eq.5 to address the numerical stability issue. This diffeomorphism is an isometry (up to a constant scaling factor) between the *Riemannian simplex* and the sphere, but NOT between the flat simplex and the sphere.
> > > >
> > > > We notice that the examples you mentioned have presumably involved the comparison of angles in different geometries (e.g., "a flat equilateral triangle with straight lines ... geodesic-boundary angle ranging [$\pi/6,\pi/2$]" assumes Euclidean geometry on Euclidean simplex, whereas the sphere angle ranging [$\pi/4,\pi/2$] assumes Riemannian geometry on sphere), which in our opinion may not be a very meaningful comparison. Instead, we tried to compare Euclidean angles between different simplexes (Riemannian vs Euclidean simplex) which are more comparable. Note **the geodesic on the Riemannian simplex are not straight lines** but are curved as demonstrated in Fig.1, therefore could make an "Euclidean angle" smaller than $\pi/6$ (i.e. more parallel to the boundary compared to "Euclidean angle" of Euclidean geodesic). If we switch to consider Riemannian angles, it should be preserved during the diffeomorphism between the sphere and Riemannian simplex, but it is not perceptually meaningful to visualize Riemannian angles, and again it makes less sense to us to compare Riemannian angles and Euclidean angles.
> > > >
> > > > As for relation to overshooting, our preliminary hypothesis is that, since modern ML optimizers and ODE solvers are mostly developed for Euclidean operators, the curved geometry having a vector field more parallel to the boundary under the "Euclidean angle" definition helps numerical stability. We have demonstrated mathematically that the direction term $\sqrt{\mu\odot\nu}-\langle\sqrt{\mu},\sqrt{\nu}\rangle \mu$, of the vector field will have a close to $u_k\approx 0$ component when $\mu_k\approx 0$, which is different from linear flow matching's fixed $\nu-\mu$. Empirically we did observe that SFM is less sensitive to ODE solver specifications in the ablation study.  Last but not least, less overshooting is only one of the potential benefits we hypothesized, and we believe that SFM's capability to employ geometry of statistical manifold and the connection to natural gradient also helps it learn efficiently in the generative model training process.
> > > >
> > > > We thank you for encouraging this insightful discussion, and we would love to continue exploring the theoretical foundations of SFM and include more such discussions in the paper.
> > > >
> > > > ## Regarding Riemannian geodesic not bounded by Euclidean
> > > >
> > > > We hope to emphasize that **our derivation in the response is consistent with our paper**, and it is not about distance on the sphere (Eq.7) but rather Riemannian geodesic (Eq.3) vs Euclidean geodesic on the simplex. To avoid confusion, we use $\mu,\nu\in\Delta^{n-1}$ as probability measures in simplex, and $x,y\in S_+^{n-1}$ as points on the sphere. To make a comparison of the different geodesics, we should use a consistent pair of points over the simplex $\mu,\nu$. In other words, we should either compare $d\_\text{cat}(\mu,\nu)$ with $\\|\mu-\nu\\|\_2,\forall \mu, \nu$, or $d\_S(\pi(\mu),\pi(\nu))$ with $\\|\mu-\nu\\|\_2,\forall \mu, \nu$, and NOT compare $d\_S(x,y)$ with $\\|x-y\\|\_2,\forall x,y\in S\_+^{n-1}$. This is the reason why our calculation was based on naive categorical distance in Eq.3 but not Eq.7 on the unit sphere. We have demonstrated mathematically that the two distance metrics are "diverse" in the sense that there does not exist a finite constant to bound the former with the latter. As we mentioned the isometric property in Proposition 1 that $d\_\text{cat}(\mu,\nu)=\frac{1}{2}d_S(\pi(\mu),\pi(\nu))$, the argument can be applied to the spherical distance (up to a constant of 2 that does not affect the bounding result).
> > > >
> > > > In short, we have provided mathematical proof that **such a fixed constant does not exist for bounding** the geodesic distance with the Euclidean distance. We are **not** trying to bound the spherical distance $d\_S(x,y)\le C\\|x-y\\|\_2,\forall x,y\in S\_+^{n-1}$, for which a fixed $C$ indeed existing between 1 and 2.

---

> > > > > ### Comment · Reviewer_WPWv · 2024-08-11
> > > > >
> > > > > Ok I see now my mistake/misunderstanding, thanks for insisting and making it clear.
> > > > > I agree that $2 d_S(\pi(\mu), \pi(\nu)) = d_{cat}(\mu,\nu)$, and that $\|\mu - \nu\|_2$ is a high distortion of $d_{cat}(\mu,\nu)$, in particular near the boundary.
> > > > >
> > > > > The point is that map $\pi$ mapping flat to curved simplex, is not bi-Lipschitz at the boundary of the domain, and even though the image is a subdomain of the sphere that is low-distortion to a simplex, the actual identification map $\pi$ isn't bounded distortion. Now I understand the overshooting comment better.
> > > > >
> > > > > I think that in order to be clear, **in the caption of figure 1 you should mention that in the upper row middle picture, do not plot the geodesics on the statistical manifold via exponential map, but rather via the parametrization map $\pi$**.
> > > > > In retrospect I now realize that the fact that you do not use the exponential map had to be evident: the exponential map of the sphere (and thus by isometry, the exp map of the statistical manifold with Fisher-Rao distance) would in fact not map it to a straight triangle as the domain, it would give a curvilinear triangle. But I didn't think twice and the "exponential" in the caption kind of made me misunderstand.

---

> > > > > > ### Author Response · Authors · 2024-08-11
> > > > > >
> > > > > > We thank your understanding and we are happy that we have reached a consensus in our discussion. We acknowledge that there might be some confusion regarding the Fig.1 caption and we will add more mathematical details to make it clearer regarding the exponential map. Again, we sincerely thank you for bringing up such an inspirational discussion, and we believe our rebuttal and clarifications have fully addressed your concerns in this regard.

---

> > > > > > > ### Comment · Reviewer_WPWv · 2024-08-11
> > > > > > >
> > > > > > > The appreciation for the thoughtful discussion is mutual, thank you for it.

---

### Official Review · Reviewer_hmQs · 2024-07-12

**Soundness:** 3
**Presentation:** 3
**Contribution:** 3
**Rating:** 6
**Confidence:** 4

**Summary:**

The paper proposes an approach to generative modelling of discrete data based on the Riemannian Flow Matching [1] algorithm where the authors use the Fisher metric to define the geometry of this space.

In detail, the authors define the generative modelling from the discrete data as sampling from the empirical distribution on the statistical manifold (the space of categorical distributions). There every point is a categorical distribution on a fixed state space and the generative model maps the noise distribution to the data distribution on the manifold, i.e. it operates with distributions of categorical distributions. In order to apply Riemannian Flow Matching, the authors define the metric tensor, tangent space, geodesics, logarithmic and exponential maps. Furthermore, for computational stability, the authors map the statistical manifold to the sphere and redefine all the differential geometric tools there.

Finally, the authors perform an extensive empirical study applying their method for language, discrete image generation, and sequence modelling in bioinformatics. The method demonstrates competitive results with prior works.

[1] Chen, Ricky TQ, and Yaron Lipman. "Riemannian flow matching on general geometries." *arXiv preprint arXiv:2302.03660* (2023).

**Strengths:**

The paper presents a complete study, i.e. it approaches an important problem in the field (generative modelling for discrete data) proposes a reasonable approach, presents the proposed approach in a comprehensible way, and provides minimal necessary empirical study of the idea.

**Weaknesses:**

Although the paper does not raise major concerns and overall satisfies the criteria of a NeurIPS publication, the practical motivation of the method is insufficient. Indeed, the problem of generating categorical distributions is interesting and the paper answers this question definitively (to a reasonable extent). However, then the authors quickly jump to the conclusion that a natural way to model discrete data is to sample a categorical distribution rather than to sample *from* a categorical distribution. Indeed, in Algorithm 2, the authors don’t even mention how they sample these categorical distributions (as I understand they consider every sample to define a point mass on the statistical manifold).

First, the transition between sampling categorical distribution (in theory) and sampling discrete data (in practice) should be stated and explained clearly. Second, this transition requires a practical motivation (perhaps an empirical study). Indeed, why would one consider the Fisher-Rao distance to be a natural distance between two sentences in a natural language? Why one would need to consider a sentence to be a categorical distribution on the statistical manifold rather than a sample from such distribution? The paper does a poor job of answering these questions and describing how the proposed method works with discrete data.

There are some minor concerns regarding the presentation:

- The name Statistical Flow Matching does not match the title of the paper.
- Line 29. It is not clear what `assumptions that fail to capitulate` mean.
- Line 52. It is not clear what `existing models often fail due to impromptu prior assumptions` means.
- Line 105 requires citations.
- In lines 112-117, please specify that equipping the manifold with the Fisher-Rao metric.
- The sampling notation in Eq. (8) is confusing due to the diffeomorphism applied to a probability density.
- There is a typo in line 118. The tangent space is called `the target space`.
- Section 3.3 reads as a preliminary section on the natural gradient rather than a connection of the proposed method to the optimization literature.

**Questions:**

I suggest adding practical examples where the data is a categorical distribution that is not concentrated in a single point. This would be a significant motivation for the proposed method.

**Limitations:**

The paper lacks a discussion of the applicability of the proposed abstraction to the generation of discrete data (see Weaknesses).

---

> ### Author Rebuttal · Authors · 2024-08-07
>
> Thank you for your high recognition of our work's significance in the realm of discrete generative modeling and for raising interesting questions and suggestions. We will fix the typo and citation in the revision. Our responses to your questions are as follows:
>
> ## Q1 Motivations
>
> ### Sampling *over* categorical distributions instead of *from* a single categorical distribution
>
> We agree that both formulations can be effectively used on discrete sampling, but their compatibility with different generative models may vary. It is important to note that SFM (and the majority of diffusion/flow models, VAEs, and GANs) are not autoregressive (AR). Unlike LM which predicts logits for the next single token deterministically given context, diffusion and flow models **operate directly on the joint distributions of all tokens**. Therefore, it is inefficient to learn a single set of deterministic categorical distributions to capture diverse joint distributions. Alternatively, the majority of non-AR methods seek to transform from prior noise distribution (e.g. multivariant Gaussians) to target joint distribution, effectively resulting in a two-step formulation where stochastic samples over joint distribution space are drawn first (i.e., logits) and then decoded with a second sampling step (argmax or softmax). Secondly, diffusion/flow models are naturally defined over continuous space. To leverage such a framework, existing discrete diffusion/flow models utilize relaxation to operate on logits which are then projected to probability simplex, similar to the statistical manifold we defined but without a properly defined geometry. In other words, the two-step sampling in SFM is not a niche design choice but rather **universal for discrete diffusion and flow models** (e.g., D3PM, SEDD, BFN, LinearFlow).
>
> Our distinction to prior works is the proposal of a better objective for training the flow model over the categorical distributions (first step), where we equip the manifold with a Riemannian metric. For the second step, we always used argmax when decoding discrete samples.
>
> ### Fisher-Rao metric
>
> We would first like to clarify a potential misunderstanding. During training, we did not use the Fisher-Rao metric for calculating the distance between two sequences (i.e., discrete samples). Instead, a metric is **necessary for defining probability paths along the geodesic** from the prior distribution (usually uniform over the simplex) towards the target distribution (concentrated around the vertex). The role of the Fisher metric is to equip the manifold of categorical distributions with a Riemannian structure on which vector fields can be calculated as the learning target. The Fisher information metric is the natural canonical Riemannian metric on the statistical manifold. It also enjoys the benefit of following the "steepest" direction of the natural gradient of decreasing KL divergence, as we have mentioned in Sec.3.3.
>
> Nevertheless, we respectfully disagree that treating discrete data as onehot distributions and using Fisher-Rao distance over them is unnatural. In fact, **discrete data have been commonly treated as distributions in classic discrete generative model training**, when using cross-entropy loss. Fisher-Rao distance is yet another metric between distributions, which indicates the steepest direction for KL divergence minimization. We will make this clear in the revision to better motivate this setup as suggested by the reviewer.
>
>
> ## Q2 Name of SFM & title
>
> As mentioned in the previous text and Sec.3, our proposed SFM is a **general generative framework for modeling measures (distributions) over the statistical manifold** of a family of parameterized distributions. In this work, we presented the realization of our framework on categorical distributions which have wide applications in various discrete generation domains. Therefore, we wish to keep the general naming of *statistical* FM and the *categorical* in the title to inform the audience about the application. We will think of a better title to avoid confusion.
>
>
> ## Q3 Impromptu assumptions in prior work
>
> For Line 29, we want to convey that the naive Euclidean assumption of the probability simplex or the logit space does not capture the true geometry of the statistical manifold. Similarly, in Line 52, we want to emphasize that the Euclidean assumption in previous existing models does not have solid mathematical grounds, which may lead to worse performance on statistical manifolds.
>
>
> ## Q4 Fisher-Rao metric
>
> The Fisher information matrix for categorical distributions is $g_{jk}(\mu)=\frac{\delta_{jk}}{\mu_j}+\frac{1}{\mu_n}$ for $1\le i,j,\le n-1$, where $\delta_{jk}$ is the Kronecker delta. Substituting this into $\langle u,v\rangle_\mu$ leads to the Riemannian inner product in Eq.4.
>
>
> ## Q5 Sampling in Eq.8
>
> We will rewrite the sampling as $x_0\sim\pi_*(p_0(\mu))$ where $\pi_*(p_0)$ denotes the standard pushforward measure induced by the diffeomorphism $\pi$. This is equivalent to first sampling $\mu_0\sim p_0(\mu)$ and then taking $x_0=\pi(\mu_0)$.
>
>
> ## Q6 Connection to natural gradient
>
> Please see the common rebuttal for details.
>
> ## Q7 Non one-hot data
>
> We really appreciate your recognition of the potential usage of SFM on general categorical distributions besides one-hot. In the paper, we have provided the Swiss Roll example as a demonstration, where other baselines with strong priors may fail. We believe that SFM could be of interest to Bayesian inference, distributional RL, and other tasks that involve sampling over distributions. We defer these use cases to our future work to extend SFM due to limited time and resources.

---

> > ### Comment · Reviewer_hmQs · 2024-08-12
> > **the motivation is still unconvincing but I consider the method to be important**
> >
> > Thank you for the response. The presentation of the proposed method as the right way to generate text remains unconvincing to me. I'm not claiming that this is the wrong approach, I feel that the paper does a poor job of convincing the reader in this. However, I recognize the significance of generative modeling on the manifold of distributions equipped with the Fisher-Rao metric (which I'm well aware of). I agree, that the Fisher-Rao metric is a fundamental concept, and that's why I think it doesn't require further motivation through the discussion of the natural gradient.

---

> > > ### Author Response · Authors · 2024-08-12
> > >
> > > We first thank your recognition of the significance of our SFM for generative modeling on the manifold of distributions and for approving the validity of our approach. We would like to further clarify why our choice of modeling discrete data as one-hot probabilities is natural and widely adopted in prior work on discrete flow/diffusion. We will make sure to improve the arrangement of sections to provide more background to the audience earlier in the text.
> > >
> > > As described in our rebuttal, previous diffusion and flow models also predominantly viewed discrete data as a distribution, and the generative task for these models effectively learns a distribution over distributions. For example, DDSM and Dirichlet FM both assumed the discrete data as probabilities following some Dirichlet distribution (points on the simplex). D3PM, MultiFlow, and SEDD viewed each token as continuous logits (points on the probability simplex after softmax). We provided extended discussions about this setting in Sec.5 in our original paper. Similarly, our SFM views tokens as points on the simplex, differing only by providing a more mathematically meaningful geometry for the probability simplex.
> > >
> > > Regarding the "right way" to generate text, to the best of our knowledge, **all current discrete diffusion and flow models for natural language modeling were based on the probability path on the simplex or related to the path on the simplex using logits**. In this sense, we believe that working with simplex is more of the "natural" and "standard" way to the diffusion community following existing work. We thank your insightful comment, which reminds us that such an assumption may diverge from that in other text modeling communities. We provided an explanation to the possible gaps and misunderstandings in our rebuttal, and we will add them to our revised manuscript to make it more accessible to general audiences with different backgrounds. We would also appreciate it if you could provide pointers to works that do not view text tokens as one-hot categorical distributions (e.g., do not use cross-entropy loss), and we will include discussions on them. Again, we sincerely thank your review and response and we hope we have fully addressed your questions and comments. We will better organize our paper in the revised manuscript.

---

### Official Review · Reviewer_SAto · 2024-07-14

**Soundness:** 3
**Presentation:** 4
**Contribution:** 4
**Rating:** 7
**Confidence:** 4

**Summary:**

This work introduces statistical flow matching, an improved method for discrete generation on the simplex by utilizing the geometry induced by the Fisher information metric. By mapping points on the simplex to points on the sphere, flows wrt the Fisher metric can be efficiently computed. SFM is tested on toy examples in the simplex as well as on binarized MNIST, Text8, and a Promotor design application.

**Strengths:**

- Great figures with above average effort and conceptual clarity that make it very easy to understand the method and theory geometrically.
- Simple, but seemingly effective addition to flow matching on data supported on a simplex.
- Could be quite significant if it is truly competitive to diffusion-based approaches.

**Weaknesses:**

- I’m quite concerned about the “Pseudo log likelihood” metric being compared to log likelihood. While the authors know and say it is incomparable, it is for some reason still compared in Tables 1 and 2 directly to the discrete NLL. I don’t understand why this is done. I might recommend an evaluation similar to MultiFlow for text8, where the evaluation is done on generated samples, and can be used to compare fully discrete, vs. the continuous state approach taken here. Its extremely concerning how much the pseudo-NLL changes based on the choice of a seemingly arbitrary threshold $t_{max}$ and the number of steps (Table 4). To me that means this score should either not be used, or should at most only be used to compare like methods.
- Qualitatively, the text8 generations look extremely poor relative to other recent related work. I’m very uncertain that “SFM achieved the second-best performance among all diffusion or flow baselines”. It would be great to include qualitative samples from other established methods for comparison. I think it is essential to establish the performance of SFM on the text8 dataset since this is arguably the only non-toy dataset and most established benchmark.

If this text8 baseline can be fixed and these results shown to hold on valid and comparable metrics, I would substantially increase my score. Even if the text8 result is not as exciting, as long as it improves over existing flow-based approaches (Including MultiFlow), I believe this paper could still be of interest to the community, and I would still consider raising my score.

MultiFlow: https://arxiv.org/pdf/2402.04997

**Questions:**

Questions:

- I don’t understand why the change of measure term is undefined at the boundary. Isn’t the transformation the square or sqrt transformation? Why is this undefined at the boundary? Could the authors elaborate on this point?
- I don’t understand the “forward noising process” $q(\tilde{\mu}_1 | \delta)$ as far as I can tell this is not defined in this work.
- Table  4,5 “ODE solver” is what solver? Euler is an “ODE solver”. Could the authors specify which solver was used?
- Why is LinearFM quite different between 300 steps and “ODE Solver” in Table 5?
- Table 5 does not look like an FID score to me, which must be strictly positive. Perhaps this is an NLL?
- Why is Multiflow cited but not compared to in the text8 task?

Minor comments:

- “we define the ODE log-likelihood as” — This is more accurately described as a change in likelihood perhaps?
- “Swill roll” —> “Swiss roll” perhaps is meant in multiple places?

**Limitations:**

Yes

---

> ### Author Rebuttal · Authors · 2024-08-07
>
> We appreciate your recognition of our work's clear presentation and novel geometric perspective for discrete generation. We'd like to clarify that Text8 results of MultiFlow were added on Jun 5 after the NeurIPS ddl, thus we were unable to compare them although it was cited.
>
> We wish to emphasize that **our major contributions are beyond the Text8 experiment**. SFM is a general generative framework for measures of categorical distributions, and we have demonstrated **the effectiveness of SFM across diverse domains** including computer vision (binarized MNIST), bioinformatics (DNA design), and NLP(Text8). We also demonstrated performance improvements **using evaluation metrics independent of likelihood calculation** (FID & Sei MSE). We further established the underlying connection of SFM to natural gradient and proposed likelihood formulation with comparability in mind.
>
> We respectfully disagree that Text8 is the only non-toy dataset. DNA design has a **tangible societal impact**, and DDSM & DirichletFM all predominantly focused on this task. The data dimension of MNIST (784) and DNA (1024) is also larger than Text8 (256). Nonetheless, we understand the importance of text8 evaluation and now include additional results in the common rebuttal following your suggestions. Below we provide more details and justifications in response to your comments.
>
> ## Q1 Choice of evaluation metrics
>
> We're afraid that you might have misinterpreted our evaluation setup. For Swiss roll and MNIST, **all baselines are diffusion or flow models with continuous parameterization**. We never compared NLLs with *discrete-space likelihood* from discrete models (e.g. autoregressive) in these tasks. Secondly, **all NLLs used in discrete tasks (Tab.1,2,4,5) are NOT pseudo-likelihoods**. They are based on the variational bond described in Eq.14 and are comparable to ELBOs from the diffusion models we compared. Only BPC for Text8 (Tab.2) was calculated based on pseudo-likelihoods as described in Appendix B.2. According to the original PLL paper, it provides a reasonable evaluation metric between non-autoregressive models.
>
> We used the **identically variational formulation** in the DDSM paper to be comparable. We also provided **FID scores that do not rely on NLL calculation**, for which our models still outperformed other baselines by a margin. More discussion on the validity of such a variational likelihood can be found in Sec.3.5 of DDSM.
>
> We justified our choice of $t_\text{max}=0.995$ in the calculation of the variational bound for NLL in Appendix D.1. We carefully chose this value to make sure it was comparable to DDSM settings. Our ablation study on the choice of $t_\text{max}$ agreed with that in DDSM, where $t$ closer to data gives a tighter estimated bound (it can't get arbitrarily low). We believe it is reasonable to compare DDSM and LinearFM using similar $t_\text{max}$.
>
> ## Q2 Additional results on Text8
>
> We provide GPT-NLL vs token entropy, and representative samples for SFM and MultiFlow with $T=0.8,0.9,1$ for visual quality check. SFM has closer to data token entropy and comparable GPT-NLL with other methods in MultiFlow's result, although slightly worse on the latter. We respectfully disagree that our samples "look extremely poor", especially when compared with MultiFlow results with matching entropy ($T=1$). The perceptual quality of SFM and MultiFlow are very close, with noticeable misspellings in both. We also found GPT-NLL very unreliable and can be easily fooled by random strings. As an example, the following sample from MultiFlow at $T=1$ *looks* even worse despite a GPT NLL of 6.648.
> ```
> she hill obhalarnnach eochrans eileann munthar cearlomha mhaonna  tardare mho mord tore lian linn mu phaile gael cangallauig laothuis guilleith leois glion guildh lara gall innonte tilbonne guilecht shuachtansert guillaste guatnaoic asthache cuichant conai
> ```
>  **We will include these results on Text8 for a more comprehensive evaluation as you suggested, and tone down on our original claims. We sincerely hope that our contributions besides text8 can be valued properly.**
> ## Q3 Change of measure term
> The change of measure term describes the change of the log-likelihood between different manifolds as $\log p_S(x)=\log p_\Delta(\mu)+\log |\det d\pi(\mu)|$ where $x=\sqrt{\mu}$. Although the density is always well-defined, the log-likelihood is undefined at the boundary, as it involves taking the logarithm of a zero probability density (see Eq.37 & 38).
>
> ## Q4 Forward probability $q(\mu|\delta)$
>
> The detailed definition of the forward probability $q(\mu|\delta)$ was provided in Eq.29 & 30 in Appendix B.1. The forward diffusion process $q_t$ defines a small neighborhood for variational estimation. In DDSM, the authors followed the fixed probability path of the Jacobi diffusion process with known forward probability. In our flow-based setting, we can use simpler indicator measures as the linear interpolation between the delta measure and the uniform distribution on the simplex.
>
>
>
> ## Q5 ODE solver
>
> As described in Sec.4.2, all generated results and NLL calculations are based on the Dopri5 solver unless otherwise specified. The Dopri5 solver is an adaptive solver with a good balance between accuracy and efficiency.
>
>
> ## Q6 LinearFM Sensitivity to solvers
>
> We noted that LinearFM tends to have very negative divergence around the one-hot distribution. Therefore, Euler step may overestimate the contribution of divergence in this case while adaptive solvers can provide a more accurate result. In contrast, as the SFM vector field is defined on the sphere, it is less sensible to solvers and results in a more stable divergence.
>
>
> ## Q7 Typos & minor issues
>
> The results in Tab.5 are NLLs. The ODE likelihood defined in Eq.12 is indeed the change of log-likelihood: $\log p^\text{ODE}:=\log p(\mu_1)-\log p(\mu_0)$. We thank you for pointing out these issues, and we will fix them in the revised manuscript.

---

> > ### Author Response · Authors · 2024-08-11
> > **Alternative BPC Calculation**
> >
> > We hope our previous rebuttal has adequately addressed your questions. To further support our discussion, we would like to introduce a potentially more comparable BPC formulation inspired by a new concurrent work MDLM [1]. MDLM provided a general form of ELBO for the continuous flow setting with a more formal proof. This ELBO applies to both continuous flow models and flow models that rely on discrete jumps with logits. Specifically, the ELBO has the form:
> > $$
> > \mathcal{L}=-\mathbb{E}_{\mu_1\sim q(\mu)}\left[\int_0^1\frac{1}{1-t}\log\langle \mu_t,\mu_1\rangle dt\right]
> > $$
> > where $\mu_t$ follows the predicted inverse trajectory starting from $\mu_1$. We believe BPC calculated with this ELBO is comparable to SEDD/D3PM/Multiflow (as they were compared in [1] as well). The BPC based on this variational bound is provided in the following table. Our BPCs were averaged over the first 1000 sequences of length 256 in the test set.
> >
> > | Model          | BPC↓          |
> > | -------------- | ------------- |
> > | SFM w/ OT      | 1.412 ± 0.006 |
> > | SFM w/o OT     | 1.410 ± 0.004 |
> > | LinearFM       | 2.197 ± 0.008 |
> > | D3PM-absorb    | 1.45          |
> > | D3PM-uniform   | 1.61          |
> > | BFN            | 1.41          |
> > | SEDD-absorb    | 1.32          |
> > | SEDD-uniform   | 1.41          |
> > | Discrete Flow  | 1.23          |
> > | Argmax Flow    | 1.80          |
> > | MultiFlow η=0  | 1.41          |
> > | Transformer XL | 1.08          |
> >
> > Similar to our evaluation results using GPT-J-6B, **our SFM reached a comparable BPC with MultiFlow, BFN and SEDD-uniform, outperformed many existing discrete diffusion/flow models including D3PM, LinearFM, and Argmax flow**. We will update the results in the revised manuscript as such BPCs are more comparable than pseudo-BPCs.
> >
> > We sincerely hope that our discussion addressed all your concerns and we look forward to further discussion.
> >
> > [1] Sahoo, Subham Sekhar, et al. "Simple and Effective Masked Diffusion Language Models." arXiv preprint arXiv:2406.07524 (2024).

---

> > > ### Comment · Reviewer_SAto · 2024-08-12
> > >
> > > I thank the authors for their extensive additional evaluations on text8 and detailed responses. I believe the evaluations are substantially improved and give a much clearer picture of the performance of various methods. I particularly appreciate the inclusion of the new ELBO from the concurrent MDLM work. I believe this is much better than what was suggested in MultiFlow, which has its problems that the authors have pointed out. While SFM is not the best method for this task, and probably not where the community will settle given other concurrent work. I am satisfied with its performance and appreciate the contribution overall. I thank the authors for their hard work and raise my score 3 --> 7.

---

> > > > ### Author Response · Authors · 2024-08-12
> > > >
> > > > We thank your recognition of our work and we are happy that our rebuttal has fully addressed your questions. Again, we thank your insightful review regarding the evaluation of the Text8 dataset that helps make our work more concrete and comprehensive. We will make sure to include these new evaluation results of both the GPT-J-6B NLLs and the more solid and comparable BPCs from the MDLM paper in our revised manuscript.

---

### Official Review · Reviewer_xYE5 · 2024-07-15

**Soundness:** 4
**Presentation:** 4
**Contribution:** 3
**Rating:** 7
**Confidence:** 4

**Summary:**

This paper proposes flow matching on the manifold of discrete distributions, where each point represents a probability mass function (pmf). In essence, this is done by parameterizing a vector of size n for each n-class categorical distribution. Instead of using Euclidean geodesic,s the authors discuss the use of Riemannian geodesics given the Fisher information matrix. To address the constraint that this should represent a probability mass function, the authors suggest using constraint ||x||^2 = 1 (i.e. the normalization is over L2), which allows geodesics to be well-defined on when points lie on the boundary. Furthermore, since this is within a continuous normalizing flow framework, authors claim that it is possible to compute exact log-likelihoods, as opposed to related diffusion model counterparts. Experiments are carried out over a range of datasets where n=2, 26, 4.

**Strengths:**

- Well written. I found the writing clear and to the point.
- A natural extension of existing works to flow matching on the manifold of discrete pmfs.

**Weaknesses:**

Overall, I feel the idea is a straightforward adoption of prior work on riemannian FM in terms of novelty, but the paper is nicely presented and packaged together. The topic of generative flows with discrete data is also timely.

**Questions:**

### Main questions / concerns:

_Regarding likelihood definition._ In Sec 2.1, you use Radon-Nikodym derivative $p=\frac{d\mu}{d\nu}$ to define the density p, where $\nu$ is used to denote the reference measure. In Sec 3.5, you state the change of measure as if it is a normal density function rather than treat it as a Radon-Nikodym derivative.
- How do you chose $\nu$?
- Can you say that the Radon-Nikodym derivative is defined for all $\mu$ that you use in practice?
- Does this change of measure (Eq 11) correspond to some $\frac{d\mu_t}{d\nu}$, i.e., fixed reference measure?

I feel that even though $p$ is defined through a Radon-Nikodym derivative, there isn't actually a formal treatment of $p$. In the text, $p$ seems to just be treated as a density function. This should be okay specifically because $x$ lies in a finite dimensional space, but this special case of the Radon-Nikodym derivative should be stated.

_Regarding choice of metric in likelihood computation._ In Tables 1 & 2 involving NLL values, are you sure these values are comparable between methods? For instance, the NLL depends on the choice of metric g (as in Eq 11), so if the $div_g$ is replaced with Euclidean divergence, this will have very different NLL values. I see that in both tables, LinearFM is directly compared to SFM. Even if different g's are used during training, they can still be compared if using the same g for NLL computation.
- Did you use the same g for NLL computation?
- Which g did you use (Euclidean or Riemannian)?

_Regarding choice of metric in desiging conditional u_t._ In Figure 1, it might be a good idea to also visualize the velocity field u_t. I think there's also an interesting property of the Riemannian u_t that isn't discussed, and it's that this u_t is parallel to the boundary. Whereas the Euclidean u_t will have a nonzero inner product between the u_t and the normal direction when evaluated on / close to the boundary. Is this true?

_Regarding the connection to natural gradient._ It's unclear what the point of Section 3.3 is. Right now, this section feels pointless as it doesn't say anything about the proposed method. What your point is should be made more explicit.
- Why should we care if the Fisher information matrix shows up as the Hessian of KL? What does this imply?
- Can you show that the geodesic under this Riemannian metric is following the "natural gradient"? Is this what you wanted to claim here?


### Minor suggestions:

- State in the main text that you are using minibatch OT couplings. Sec 3.4 does not mention how OT is used in the actual training algorithm. (I had assumed it is done over the full training set until finding Alg 2.)
- The "target" space --> "tangent" in Sec 3.1.

**Limitations:**

Authors adequately addressed limitations.

---

> ### Author Rebuttal · Authors · 2024-08-07
>
> Thank you for your appreciation of our work's clear presentation as a natural and timely extension of Riemannian FM on discrete data. We thank your thoughtful suggestions and will fix typos and clarify minibatch-OT in the revision. We'd like to address your comments as follows.
>
> ## Q1 Regarding likelihood definition
>
> We mentioned the Radon-Nikodym derivative in 2.1 as the definition of probability density functions on general manifolds. In our likelihood derivation (Sec.3.5), the density $\frac{dP}{dV}$ can be well-defined over the statistical manifold which is finite-dimensional ($n-1$ dimensional for $n$ categories) using the Riemannian volume as the reference measure. This is in contrast to works like functional FM where data points are continuous functions lying on the infinite-dimensional Hilbert space. We will clarify this as you suggested.
>
> In terms of the choice of measure, we make sure to use a consistent reference measure for all change-of-measures (COM) in Sec 3.5. Note that the Riemannian volume on the sphere coincides with the Euclidean one (see response to Q2), and only the COM for the two transformations $\pi$,$\pi^{-1}$ will be impacted by the choice of measure. To make the NLL comparable to Euclidean flows, we have kept the reference measure consistent as the Euclidean volume form in the equations in Appendix B.1. We will make these mathematical details clearer and explicitly state the choice of volume in the revised manuscript.
>
> For categorical distributions $\mu$, we used the canonical counting measure over the discrete sample space $\mathcal{X}=\{0,1,\dots,n\}$.
>
>
> ## Q2 Regarding likelihood computation
>
> We thank your insightful question regarding the divergence in NLL calculation. You are completely right that even though vector field is learned with Riemannian geometry, we can still compute likelihood with Euclidean divergence. As a matter of fact, we have taken comparability into consideration in our derivation of likelihood. Note that for manifolds that can be embedded in the ambient Euclidean space (e.g., simplex and sphere), the Riemannian divergence is the same as Euclidean divergence (for embeddable manifolds, we have $\log|\det g|\equiv0$). By using the Euclidean volume form as the reference measure in all COMs and computing prior likelihood on Euclidean simplex, **we effectively used Euclidean divergence on the simplex** for SFM and LinearFM to guarantee that the results are comparable.
>
> It is possible, though, to calculate the Riemannian likelihood for SFM. To do this, we need to make several adaptations. The prior likelihood should be adjusted to $p_0=\Gamma(n)/\sqrt{|\det g|}$ such that the integral $\int_\Delta p_0dV=\int_\Omega p_0\sqrt{|\det g|}d\mu_1\cdots d\mu_{n-1}=1$. The change of measure term in Eq.37 & 38 also needs to be adjusted as $\log|\det d\pi|=(n-1)\log 2$ and $\log|\det d\pi^{-1}|=-(n-1)\log 2$.
>
>
> ## Q3 Regarding choice of metric in designing conditional $u_t$
>
> Thank you for mentioning the nice property of Riemannian $u_t$ being parallel to the boundary. We'd like to mention that our visualization of the logarithm map can be viewed as a demonstration of $u_t$'s direction  (up to a constant scaling factor that depends on $t$), as the vector field can be calculated in terms of the logarithm map as $u_t(\mu_t|\mu_0,\mu_1)=\log_{\mu_t}(\mu_1)/(1-t)$ (assuming a linearly decreasing geodesic distance). We indeed noticed that the Riemannian structure induced by the Fisher information metric leads to vector fields more parallel to the boundaries. This can also be demonstrated mathematically by looking at the logarithm map in Eq.23. Consider the direction term $\sqrt{\mu\odot\nu}-\langle\sqrt{\mu},\sqrt{\nu}\rangle \mu$. For $\mu$ close to the boundary with $\mu_k\approx 0$, its corresponding vector field will also have a close to $u_k\approx 0$ component, which is different from linear flow matching's fixed $\nu-\mu$.
>
> We hypothesize that one potential benefit of such a curved geometry over the naive Euclidean geometry is that the former helps **prevent overshooting across the boundaries**. Specifically, consider a target point on the boundary. The Euclidean vector field will continue to push the points outside the manifold, whereas the Riemannian vector field tends to travel parallel to that boundary to prevent going across the manifold. The sphere exponential map also naturally ensures the point (after transformation back) lies on the simplex. Once overshooting happens during sampling, the model may exhibit undefined behavior as it was never trained on points outside the manifold. We will emphasize this benefit more clearly in our future revision.
>
>
> ## Q4 Regarding the connection to natural gradient
>
> Please see the common rebuttal for details.

---

### Author Rebuttal · Authors · 2024-08-07

Dear all Reviewers,


We sincerely appreciate your reviews that help make our work more concrete and comprehensive. We thank you for recognizing our novel geometric perspective of discrete generation and our paper's clear presentation. Here we address some of the common questions.


## More details regarding connection to natural gradient

As per request by Reviewer xYE5, WPWv, and hmQs, we elaborate more on the connection to natural gradient in addition to Sec.3.3. We want to demonstrate **the direction of the Riemannian vector field induced by the Fisher information metric coincides with the steepest direction of minimizing local KL divergence** (the natural gradient).

**From the optimization view point**, one objective for generative modeling of categorical distributions would be the KL divergence $D_\text{KL}(\tilde{\mu}\\|\mu)$ between the generated distribution $\tilde{\mu}$ and the target distribution $\mu$. The fact that the Fisher information metric is the Hessian of KL divergence allows us to locally expand the change of KL divergence in Eq.10 in terms of Fisher information metric as $D_\text{KL}(\mu(\theta)\\|\mu(\theta_1))\approx \frac{1}{2}\sum_{jk}\Delta\theta_j\Delta\theta_j g_{jk}=\frac{1}{2}\\|\Delta\theta\\|^2_g$ where $\Delta\theta=\theta-\theta_1$. The steepest direction $\Delta\theta$ decreasing the KL divergence is known as the *natural gradient* in the existing literature.

**From the geometric view point**, the geodesic, by definition, is a (locally) length-minimizing curve with respect to the corresponding Riemannian metric. Therefore, by following the direction of the vector field that decreases the geodesic element $ds^2=\\|d\theta\\|^2_g$, we are indeed following the steepest direction of natural gradient that minimizes the local KL divergence. This theoretical connection may contribute to the better performance of SFM.

## Additional results for Text8

We first noted that, though we cited MultiFlow, their Text8 experiment was not added until Jun 5 (>2 weeks after NeurIPS ddl). Nevertheless, as requested by Rewiewer SAto, we now add results with the GPT-J-6B model in addition to BPC, following the setting in MultiFlow. We emphasize that **pseudo-LL is only used in Text8 BPC** and all discrete NLL comparisons follow the similar ELBO derivation in DDSM. We only compared NLLs to diffusion/flow models with comparable ELBO formulations (see the response to SAto) and used consistent divergence (see the response to xYE5). We also provided **non-NLL metrics** (FID, Sei MSE) for MNIST/DNA data which are **non-toy high-dimensional real-world datasets**, where SFM outperformed all baselines with a margin. We sincerely hope that these contributions can still be valued.

As GPT-J-6B was not trained on Text8, its NLL does not necessarily reflect Text8 distribution fitness. From the MultiFlow results, such an NLL can be **made artificially low by duplicating high-frequency words e.g. numbers**, thus joint comparison of GPT NLL and token entropy was used, where an entropy closer to data is preferred. We show in the table below that SFM tends to produce samples with data entropy closer to data, and the GPT NLLs are still comparable to (slightly worse) MultiFlow/D3PM when entropy is close. As SFM directly samples categorical probabilities instead of making multiple discrete jumps with logits, it is incompatible with temperature-tuning. Thus in the table below, we compare with $T=1$ which matches the entropy of SFM. We also noted that GPT NLL can be **easily fooled by randomly generated strings** based on letter frequency. Additionally, low-T MultiFlow variants achieved lower NLLs than the ground truth data, making this metric less credible. We also noticed that the **low NLL samples for MultiFlow often consist of repeated numbers with little semantic meaning**. To enable fair comparison, we provide multiple generated samples (above/below/just-on-average NLL for each model) and their GPT NLLs in the PDF. Different temperature settings for MultiFlow are also included using checkpoint provided on Github. We noted the big impact of temperature on MultiFlow as it generated more numbers repeatedly with lower temperature. In contrast, SFM achieved perceptually similar or even better results with more diverse vocabulary.

We thank the authors of MultiFlow for generously providing their raw data in Fig.2. The complete figure is provided in the pdf.

| Model  | GPT-J-6B score ↓ | Token entropy ↑ |
| - | - | - |
| Data | 4.099 | 7.479 |
| Random | **5.827** | 5.519 |
| SFM w/ OT | 7.071 | 7.347 |
| SFM w/o OT | 7.154 | **7.388** |
| LinearFM | 7.490 | 7.118 |
| SEDD mask | 6.49 | 7.17 |
| SEDD absorb | 6.28 | 6.97 |
| MultiFlow | 6.728 | 7.387 |
| MultiFlow η = 0 | 6.729 | 7.325 |
| D3PM | 6.929 | 7.379 |
| Autoregressive | 6.296 | 7.165 |


We thank Reviewer SAto for proposing an alternative evaluation. We will include these results on Text8 for more comprehensive evaluation, and tone down on our original claims. Nevertheless, we'd like to note that the **general formulation of SFM over continuous statistical manifold has potential beyond NLP discrete sampling**, and could be of interest to Bayesian inference, distributional RL, and other tasks that involve sampling over distributions.

NLL used to be one of the common metrics for generative models as it reflects closeness to the data distribution. It is until recently that this metric became underlooked due to arguably loose and diverse ELBO derivation in diffusion, and researchers resort to third-party models (InceptionV3, GPT, AF2, etc.) fitted on different data that may be subject to artifact and bias. We'd like to call for better community efforts on this issue. We provided exact NLL and a tight ELBO formulation with a goodwill that this could be a step towards better evaluation of flow models, and we have tried our best to make it as comparable as possible.

---

### Author Response · Authors · 2024-08-14
**Sincere Gratitude from Authors**

Dear Reviewers,

We sincerely thank you for your insightful and constructive feedback throughout the review process. Your thoughtful comments have helped make our work more concrete and comprehensive. We are glad that our rebuttals have successfully addressed your questions and comments, and we are grateful for your unanimous recognition of our efforts and contributions in our proposed SFM framework.

We especially enjoyed the interactive discussions during the rebuttal period, which has been inspirational and fruitful for us. We will make sure to add the additional evaluation metrics on Text8, clarify some mathematical notations and details, and include additional insights inspired by our discussions in our revised manuscript. Again, we thank you for your valuable reviews and for helping us bring this research to fruition.

Warm Regards,
Authors.

---

### Decision · Program_Chairs · 2024-09-25

**Decision:**

Accept (poster)

**Comment:**

The paper extends flow matching to discrete data by working with the associated simplex equipped with the usual Fisher-Rao metric. This is a natural, yet novel, usage of Riemannian flow matching to statistical manifolds. Working with generative flows for discrete data is timely and the paper makes a sound contribution to the field.